# MASTARS: Multi-Agent Sequential Trajectory Augmentation with Return-Conditioned Subgoals

## Abstract

The performance of offline reinforcement learning (RL) critically depends on the quality and diversity of the offline dataset. While diffusion-based data augmentation for offline RL has shown promise in single-agent settings, its extension to multi-agent systems poses challenges due to the combinatorial complexity of joint modeling and the lack of inter-agent coordination in independent generation. To overcome these issues, we introduce MASTARS, a novel diffusion-based framework that generates coordinated multi-agent trajectories through agent-wise sequential generation. MASTARS employs a diffusion inpainting mechanism, where each agent's trajectory is generated based on the trajectories of previously sampled agents. This enables fine-grained coordination among agents while avoiding the complexity of high-dimensional joint modeling. To further improve sample quality, MASTARS incorporates return-conditioned subgoals, allowing it to leverage valuable data that might otherwise be discarded. This agent-wise, goal-conditioned approach produces realistic and harmonized multi-agent rollouts, facilitating more effective offline MARL training. Experiments on benchmark environments demonstrate that MASTARS significantly improves the performance of offline MARL algorithms.

## 1 Introduction

Offline reinforcement learning (RL) (Levine et al., 2020) aims to learn policies from pre-collected datasets without additional environment interaction, enabling safe and efficient policy development in domains where online exploration is costly or risky. Recent works in this area have achieved remarkable success by constraining the policy to remain close to the behavior policy (Fujimoto et al., 2019; Fujimoto & Gu, 2021; Kostrikov et al., 2022; Tarasov et al., 2023) or by regularizing value functions through penalizing values for out-of-distribution (OOD) actions (Kumar et al., 2020; Lyu et al., 2022; Mao et al., 2023). However, the effectiveness of current offline RL approaches is significantly influenced by the quality and diversity of the dataset (Schweighofer et al., 2022). To overcome this limitation, *data augmentation* has emerged as an important research topic in offline RL. In particular, diffusion models are useful for this purpose and many data-augmentation methods using diffusion models (Janner et al., 2022; Liang et al., 2023; Lu et al., 2023; He et al., 2023; Li et al., 2024; Lee et al., 2024) have been proposed. These methods generate additional data resembling the offline dataset distribution by leveraging the strong generative power of diffusion models (Ho et al., 2020), which have shown remarkable performance in the image domain.

Extending offline single-agent RL data augmentation to multi-agent RL (MARL) introduces challenges due to the combinatorial growth of joint state–action spaces and the necessity for coordinated behavior. Prior works (Yang et al., 2021; Pan et al., 2022; Wang et al., 2023a; Shao et al., 2023; Kim & Sycara, 2025) propose specialized approaches tailored to the multi-agent setting to recover coordinated policies in offline MARL scenarios, highlighting essential distinctions from the single-agent setting. Although these works address the offline multi-agent RL problem, they still face limitations similar to offline single-agent RL, with performance heavily influenced by the *quality* and *diversity* of the dataset. In contrast to single-agent augmentation methods that model distributions of individual trajectories, multi-agent settings require capturing coherent joint behavior across agents. Independent per-agent diffusion models can hinder coordination by ignoring cross-agent dependen-

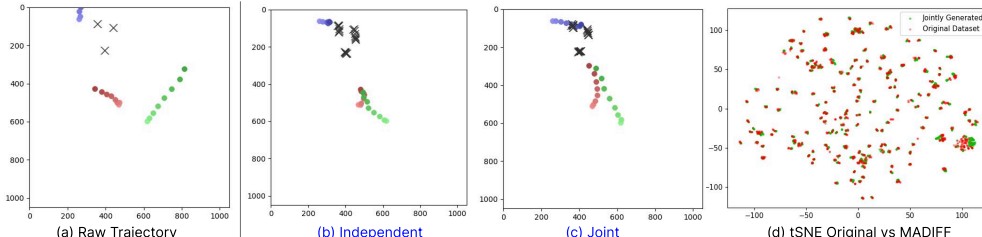

Figure 1: (a) Raw trajectory, (b) independent generation, (c) Joint generation, and (d) t-SNE of original dataset (red) and jointly generated dataset (green)

cies, while naively concatenating all agents' data into a single model for joint generation often leads to undesirable generation and sample inefficiency due to high dimensionality.

Fig. 1 shows one such example, showing the results of agent-wise independent generation and joint generation, both based on MADiff(Zhu et al., 2024)*[1],an existing diffusion model-based data augmentation method . (A detailed explanation of this generation is provided in Appendix B.) The considered offline task is the Spread medium-replay task from the Multi-agent Particle Environment (MPE) (Lowe et al., 2017a), where each of three agents (green, red, and blue) must reach one distinct landmark out of total three (marked 'X') without collisions. As shown, neither the independent nor the joint strategies easily produces a fully cooperative trajectory. The independent approach frequently results in collisions as agents ignore each other's behavior. In contrast, the joint method improves collision avoidance but still suffers from suboptimal coordination mainly due to the suboptimal quality (medium-replay) of the dataset: Note that both red and green agents converge on the same landmark in Fig. 1. In addition to unsatisfactory coordinated behavior generation, direct use of diffusion models for joint generation significantly limits the diversity of generated trajectories. Fig. 1 (d) shows the t-SNE plot of the generated (green) and original (red) trajectories of three agents' observations and actions (each 1000 trajectories) into 2D. As shown, the t-SNE projections of the generated trajectories mostly overlap with those of the original trajectories in the dataset, i.e., well following behavior cloning. These observations hint that diverse coordinated-behavior trajectories are not easily generated. To this end, several works (Zhu et al., 2024; Oh et al., 2024; Yuan et al., 2025; Li et al., 2025; Fu et al.) propose joint dataset generation methods for multi-agent systems by incorporating attention-based diffusion, global Q-total functions, or trajectory stitching. However, these methods generate entire multi-agent trajectories jointly, limiting their ability to capture fine-grained relational dependencies among agents.

Then, the key question is "*how can one generate diverse coordinated trajectories even better than the original trajectories?*" To address this question, we propose Multi-Agent Sequential Trajectory Augmentation with Return-Conditioned Subgoals (MASTARS), a novel diffusion-based framework that generates cooperative multi-agent trajectories in an agent-wise sequential manner guided by high-return subgoals. MASTARS is built upon three core components: (i) partial generation based on subgoal identification, which determines the segment to be generated while keeping the good portion of the original trajectory for each agent by leveraging value estimates, (ii) agent-wise sequential trajectory generation strategy, which generates each agent's trajectory in coordination with the others through the key idea of *inpainting*, and (iii) value-based generation ordering, which determines the order of agent-wise sequential generation based on value estimates.

MASTARS ensures coordinated data augmentation in the multi-agent setting, enhancing the overall performance of any offline MARL method when applied to the augmented dataset. Experiments conducted on various MARL benchmarks such as the Multi-Agent Particle Environments (MPE) (Lowe et al., 2017a) and the StarCraft Multi-Agent Challenge (SMAC) (Samvelyan et al., 2019a) with the off-the-grid MARL (og-marl) dataset (Formanek et al., 2023) demonstrate that the dataset augmented by MASTARS significantly improves the performance of various offline MARL methods (Shao et al., 2023; Pan et al., 2022). The main contributions of this work are:

- A novel framework for agent-wise sequential data augmentation with return-conditioned subgoals, which efficiently generates coordinated trajectories across agents.
- Demonstration of the effectiveness of generating harmonized multi-agent trajectories using an inpainting-based approach within a MARL data augmentation framework.

---

[1]MADiff is an offline MARL method; we use its trajectory generation component, denoted with asterisk (*).

- Empirical validation on various tasks (Lowe et al., 2017a; Samvelyan et al., 2019a), showing substantial performance gains across several offline MARL methods (Shao et al., 2023; Pan et al., 2022), along with a detailed analysis of the generated trajectories.

## 2 PRELIMINARIES

### 2.1 MULTI-AGENT OFFLINE REINFORCEMENT LEARNING

We consider a cooperative Partially Observable Markov Game with $N$ agents, formalized as a Dec-POMDP (Oliehoek & Amato, 2016) $\langle N, \mathcal{S}, \{\mathcal{A}^i\}_{i=1}^N, \Omega, \mathcal{O}, P, r, \rho, \gamma \rangle$, where $\mathcal{S}$ is the state space, $\mathcal{A}^i$ is the action space of agent $i$, $\Omega$ is the observation space, $\rho : \mathcal{S} \to [0, 1]$ is the distribution over initial states and $\gamma \in [0, 1)$ is the discount factor. The agents share the environment state $s \in \mathcal{S}$ but receive private observations $\{o_i \in \Omega\}$ according to the observation function $\mathcal{O}(s, i) :$ $\mathcal{S} \times \mathcal{N} \to \Omega$ for each agent $i$ at each timestep. The agents select actions $a^i \in \mathcal{A}^i, \forall i$, forming the joint action $\mathbf{a} = (a^1, \ldots, a^N)$. The environment makes a transition according to dynamics $P(s' \mid s, \mathbf{a}) : \mathcal{S} \times \mathcal{A}^1 \times \cdots \times \mathcal{A}^N \to \mathbb{R}$ and returns a global reward $r(s, \mathbf{a})$. Note that we use bold symbols to denote the joint quantity of all agents, e.g., $\boldsymbol{o} = (o^1, \ldots, o^N)$.

In the offline setting, we are given a fixed dataset $\mathcal{D} = \{\tau_k^{\text{full}}\}_{k=1}^K$ of $K$ trajectories $\tau^{\text{full}} = (\boldsymbol{o}_{0:T}, \boldsymbol{a}_{0:T-1}, r_{0:T-1})$ collected by an unknown behavior policy. No further interaction with the environment is allowed. The goal is to learn a decentralized policy set $\boldsymbol{\pi} = \{\pi^i(a^i \mid o^i; \theta^i)\}_{i=1}^N$ (or an equivalent centralized-training factorization) that maximizes the expected return $J(\boldsymbol{\pi}) = \mathbb{E}_{\tau \sim \boldsymbol{\pi}, P}\Big[ \sum_{t=0}^{T-1} \gamma^t r_t \Big]$.

### 2.2 DIFFUSION-BASED GENERATIVE MODELS

Diffusion models (Ho et al., 2020; Song et al., 2021) consist of a forward process that gradually corrupts data with Gaussian noise and a learned reverse process that denoises the corrupted data. In the forward process, noise is injected into $x_0 \sim q(x_0)$ over $K$ steps using a fixed variance schedule $\{\beta_k\}_{k=1}^K$: $q(x_k \mid x_{k-1}) = \mathcal{N}(\sqrt{1 - \beta_k} x_{k-1}, \beta_k I)$, resulting in a Gauss–Markov process with $q(x_k \mid x_0) = \mathcal{N}(\sqrt{\bar{\alpha}_k} x_0, (1 - \bar{\alpha}_k)I)$, where $\bar{\alpha}_k = \prod_{i=1}^k (1 - \beta_i)$. The posterior $q(x_{k-1} \mid x_k, x_0)$ is also Gaussian. The generative model $p_\theta$ starts from $x_K \sim \mathcal{N}(0, I)$ and generates samples via reverse transitions: $p_\theta(x^{0:K}) = \mathcal{N}(x^K; 0, I) \prod_{k=1}^K p_\theta(x^{k-1} \mid x^k)$.

DDPM (Ho et al., 2020) converts learning $p_\theta$ to learning a noise model $\epsilon_\theta$, enabling reverse-sampling as $x_{k-1} = \frac{1}{\sqrt{\alpha_k}} \left( x_k - \frac{1 - \alpha_k}{\sqrt{1 - \bar{\alpha}_k}} \epsilon_\theta(x_k, k) \right) + \sqrt{\beta_k} z$ with $x_K, z \sim \mathcal{N}(0, I)$. A neural network for the noise model $\epsilon_\theta(x_k, k)$ is trained to predict the noise $\epsilon \sim \mathcal{N}(0, I)$ in the decomposition $x^k = \sqrt{\bar{\alpha}_k} x_0 + \sqrt{1 - \bar{\alpha}_k} \epsilon$, using the simplified MSE objective:

$$L_{\text{diff}}^{\text{DDPM}}(\theta) = \mathbb{E}_{x_0 \sim q(x_0), k \sim \mathcal{U}[1,K], \epsilon} \big[ \|\epsilon_\theta(x_k, k) - \epsilon\|^2 \big]. \tag{1}$$

**Diffusion Models in Data Augmentation** To generate offline RL datasets, diffusion models (Janner et al., 2022; Liang et al., 2023; Li et al., 2024; Lee et al., 2024) learn the trajectory distribution $p_\theta(\tau)$ via $\epsilon_\theta$ from the offline dataset using equation 1. (The basic idea of RL dataset generation with diffusion models is explained in Appendix A.) In particular, Diffuser (Ajay et al., 2023) proposed a return-conditioned diffusion model trained with the loss (Ho & Salimans, 2022):

$$\mathcal{L}(\theta) = \mathbb{E}_{k, \{\tau_0, R\} \in \mathcal{D}, \epsilon, \beta \sim Bern(p)} \Big[ \big\| \epsilon - \epsilon_\theta\big(\tau_k, k, (1 - \beta)R + \beta \emptyset\big) \big\|^2 \Big], \tag{2}$$

where $\emptyset$ is a null token for the unconditioned model. The reverse process uses the perturbed noise $(1 + w)\epsilon_\theta(\tau_k, k, R) - w\epsilon_\theta(\tau_k, k, \emptyset)$ conditioned on return $R$ with guidance weight $w$. MADiff (Zhu et al., 2024) generates sequences of observations by a return-conditioned diffusion model with an inter-agent attention architecture for MARL. We adopt MADiff as our base diffusion model and additionally train a reward function $g_\xi(\boldsymbol{o}, \boldsymbol{a})$, a transition model $f_\varphi(o^i, a^i)$, and an inverse dynamics model $I_\phi(o^i, o^{i'})$ to predict rewards, observations and actions from a generated observation sequence (Ajay et al., 2023; Li et al., 2024). Our training objective for the reward function and inverse dynamics model is defined as

$$\mathcal{L}(\xi, \phi, \varphi) = \mathbb{E}_{\boldsymbol{o}, \boldsymbol{a}, r, \boldsymbol{o'} \sim D} \Big[ \big(r - g_\xi(\boldsymbol{o}, \boldsymbol{a})\big)^2 + \sum_{i=1}^N \Big( \big(a^i - I_\phi(o^i, o^i)'\big)^2 + (o^{i'} - f_\varphi(o^i, a^i))^2 \Big) \Big].$$

$$\tag{3}$$

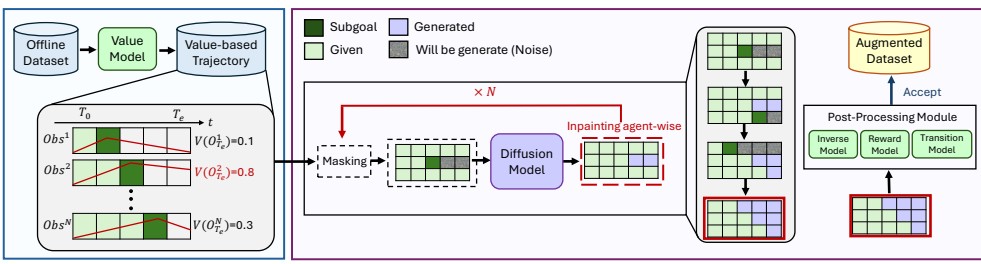

Figure 2: Overview of our trajectory generation framework. The pre-processing module (left) identifies subgoals and determines the generation order. The sequential inpainting module (center) then generates each agent's trajectory sequentially via inpainting. Finally, the post-processing module (right) completes the trajectories by generating actions and rewards and validating the trajectory.

# 3 PROPOSED METHOD

In this section we present the **M**ulti-**A**gent **S**equential **T**rajectory **A**ugmentation with **R**eturn-conditioned **S**ubgoals (MASTARS) algorithm to address the aforementioned challenges of generating cooperative datasets in offline MARL. Our method captures inter-agent correlations during trajectory sampling to enhance coordinated behavior in the augmented dataset. The MASTARS algorithm comprises three key components: a subgoal-based partial generation method, agent-wise sequential generation strategy, and a value-based generation ordering scheme.

- **Subgoal-Based Partial Generation**: MASTARS does not generate whole new trajectories from scratch but takes original trajectories from the dataset and improves them. For this, MASTARS determines a specific timestep from which generation begins, referred to as *subgoal*, while keeping the portion before the subgoal.

- **Agent-wise Sequential Trajectory Generation**: MASTARS adopts agent-wise sequential trajectory generation to ensure coordinated behavior. For this, we employ the technique of *inpainting* (Lugmayr et al., 2022) from the vision domain, which fills the masked portion of an image with best coordination with the unmasked part.

- **Value-based Agent Ordering**: MASTARS determines the agent-wise generation order so that the lately generated agent has more freedom to fit to the previously generated agents.

The trajectory generation in MASTARS relies on five pre-trained models: (1) a diffusion model $\epsilon_\theta$ for generating sequences of observations; a set of dynamics models including (2) a reward function $g_\xi$, (3) an inverse dynamic model $I_\phi$ and (4) a transition model $f_\varphi$; and finally (5) a value function $V_\zeta$. The diffusion model is trained using equation 2 with the MADiff (Zhu et al., 2024) architecture, the dynamics models are trained with equation 3, and the value function $V_\zeta$ is trained using the TD prediction. Once an observation sequence is generated, actions and rewards can be generated based on the learned models. Details of the training loss are presented in Appendix C.2.

Building on these pre-trained models, the overall flow of MASTARS is shown in Fig. 2. In the following subsections, we describe the key components of MASTARS in detail.

## 3.1 PARTIAL GENERATION VIA SUBGOAL IDENTIFICATION

To accomplish improved-quality trajectory generation, we reuse parts of existing trajectories, based on the question: "Is it always necessary to regenerate the entire trajectory from scratch?" In many cases, a trajectory may already contain segments resembling expert behavior. Regenerating the entire sequence risks discarding this valuable data. To leverage the good data segments, we incorporate a *subgoal-based generation* mechanism that identifies high-value states.

Formally, we define a MARL observation trajectory $\tau$ as an $N \times d \times H$ tensor, where $N$ is the number of agents, $d$ is the dimension of observation $o^i$, and $H$ is the time horizon. Given an original trajectory $\tau = (\boldsymbol{o}_t, \ldots, \boldsymbol{o}_{t+H-1}) \in \mathbb{R}^{N \times d \times H}$, we define *subgoal* $g^i$ for each agent $i$ as the timestep at which the agent's estimated value is highest: $g^i = \arg\max_{t' \in [t, t+H-1]} V_\zeta(o^i_{t'})$. By conditioning on this subgoal $g^i$, we selectively generate only the less optimal part of agent $i$'s existing trajectory, i.e., the portion after this highest peak point, to improve data quality while retaining useful behavior

from the original dataset under the assumption that the steps towards the highest-value state are good behavior (Jeon et al., 2022).

## 3.2 AGENT-WISE SEQUENTIAL TRAJECTORY GENERATION

To realize trajectory generation with coordinated behavior, we adopt an agent-wise sequential strategy. That is, we pick one agent and generate the trajectory segment after this agent's subgoal timestep. Then, we pick the next agent and generate the trajectory segment after the secondly-picked agent's subgoal timestep. The key point here is that at each generation step, the newly-generated trajectory segment should be well coordinated with the already given or designed segments. To enforce this, we employ the technique of *inpainting* from vision domain (Elharrouss et al., 2019). Basically, inpainting is a technique that reconstructs missing segments based on surrounding context so that the reconstructed image best fits with its surroundings. By adopting inpainting together with good portion preserving from Section 3.1 and sequential generation, we aim that the currently-generated segment well coordinates with the existing segments of good portion and previously generated agents' segments. In particular, we employ the RePaint algorithm (Lugmayr et al., 2022), a diffusion-based method that improves sample quality through iterative resampling and re-noising without extra model retraining, making it well-suited for generating coherent data.

**Inpainting-Based Agent-wise Sequential Generation.** Given an initial trajectory $\tau^0 = (\boldsymbol{o}_t, \ldots, \boldsymbol{o}_{t+H-1}) \in \mathbb{R}^{N \times d \times H}$ and a generation order $(\kappa(1), \ldots, \kappa(N))$, we reconstruct each agent's trajectory using the RePaint algorithm conditioned on other agents' observations. We therefore invoke RePaint $N$ times, once for each agent, to complete the full trajectory generation; these $N$ invocations form the *outer loop*, while the denoising steps inside each RePaint outer loop constitute the *inner loop*. Note that we take $\tau^0$ directly from the dataset and refine it, as already mentioned in Section 3.1.

At the $l$-th RePaint generation, we start with $\tau^{l-1}$, the partially generated trajectory after the previous $l-1$ generation steps; that is, the trajectories for agents $\kappa(1), \ldots, \kappa(l-1)$ have already been generated. To distinguish the portion under current generation, we define a binary mask $m^l$ of size $N \times d \times H$

$$m^l(j, d, t) = \begin{cases} 1 & \forall d, \quad \text{if } j = \kappa(l) \text{ and } t > g^j \\ 0 & \forall t, d, \quad \text{otherwise} \end{cases} \tag{4}$$

where only the $\kappa(l)$-th agent segment after subgoal timestep $g^j$ is set to 1.

With the binary mask $m^l$, we initiate generation. The generating reverse sampling inner loop starts with index $k = K$, where $K$ is the diffusion step from equation 1. For $k = K$, we initialize $\tilde{\tau}_K^{\text{current-agent}} \sim \mathcal{N}(0, I)$ and $\tilde{\tau}_K^{\text{others}} \sim \mathcal{N}(\sqrt{\bar{\alpha}_K}\tau^{l-1}, (1 - \bar{\alpha}_K)\mathbf{I})$. Then, the denoising steps are applied with the following sampling rules from $k = K$ to $k = 1$ to obtain $\tilde{\tau}_0^l$:

$$\tilde{\tau}_k^l = m^l \odot \tilde{\tau}_k^{\text{current-agent}} + (1 - m^l) \odot \tilde{\tau}_k^{\text{others}} \quad \in \mathbb{R}^{N \times d \times H} \tag{5a}$$

$$\tilde{\tau}_{k-1}^{\text{current-agent}} = \frac{1}{\sqrt{\alpha_k}} \left( \tilde{\tau}_k^l - \frac{1 - \alpha_k}{\sqrt{1 - \bar{\alpha}_k}} \epsilon_\theta(\tilde{\tau}_k^l, k) \right) + \sqrt{\beta_k} z \quad \text{where} \quad z \sim \mathcal{N}(0, I) \tag{5b}$$

$$\tilde{\tau}_{k-1}^{\text{others}} \sim \mathcal{N}\left(\sqrt{\bar{\alpha}_k}\tau^{l-1}, (1 - \bar{\alpha}_k)\mathbf{I}\right). \tag{5c}$$

Note that the input for others is a perturbed version of $\tau^{l-1}$ for $k > 1$ because $\bar{\alpha}_k < 1$ for $k > 1$. This improves the coherence between the current generation and other parts. Note also that throughout this reverse sampling internal loop for generating data for agent $\kappa(l)$, other agents' existing data are appended for coordinated generation as seen in equation 5a. After completing the denoising internal loop, the final output of the $l$-th trajectory generation is obtained as $\tau^l = \tilde{\tau}_0^l$, representing the updated trajectory in which current agent $\kappa(l)$'s trajectory has been generated, while all other agents' trajectories remain unchanged. Note that $\bar{\alpha}_1 = 1$ and hence equation 5c generates $\tilde{\tau}_0^{\text{others}} = \tau^{l-1}$ at the final step $k = 1$.

This inpainting procedure is repeated in order of $(\kappa(1), \ldots, \kappa(N))$ until all agent trajectories are generated.

**Post-Processing and Validation for Generated Dataset.** At the end of the above sequential generation, we obtain a newly generated observation trajectory $\tau = (\tilde{\boldsymbol{o}}_{t:t+H-1})$, where the tilde represents generated quantities. To augment this trajectory into an offline dataset, we must generate

the corresponding action and reward sequences using the pre-trained inverse dynamic model $I_\phi$ and reward model $g_\xi$, resulting in the following predictions: $\tilde{a}_{t'}^i = I_\phi(\tilde{o}_{t'}^i, \tilde{o}_{t'+1}^i)$ and $\tilde{r}_{t'} = g_\xi(\tilde{\boldsymbol{o}}_{t'}, \tilde{\boldsymbol{a}}_{t'})$

Finally, to decide whether a generated trajectory is suitable for augmentation, we assess its consistency with the environment dynamics using a pre-trained transition model $f_\varphi$. Trajectories that deviate too much from the original distribution may cause out-of-distribution (OOD) errors. Thus, we include a trajectory in the augmented dataset only if the following condition is satisfied (Li et al., 2024):

$$\mathbb{E}_{i\in[1,N],\ t'\in[t,\,t+H-1]}\left\| \tilde{o}_{t'+1}^i - f_\varphi(\tilde{o}_{t'}^i, \tilde{a}_{t'}^i)\right\| < \epsilon \tag{6}$$

where $\epsilon$ is a threshold controlling the acceptable prediction error under the transition model.

### 3.3 Generation Ordering via Value and Subgoals

The performance of the aforementioned sequential generation depends on the generation order $(\kappa(1),\ldots,\kappa(N))$. Now, we provide our criterion for determining this generation order by prioritizing the agents who can contribute most to coordinated behavior trajectory generation.

**Order of Generation.** We improve coordinated and high-quality trajectory generation by ordering agents according to their importance during future planning. Specifically, we propose two complementary ordering schemes.

The first scheme prioritizes agents whose current situations are expected to have greater future returns. Given a trajectory $\tau = (\boldsymbol{o}_{t:t+H-1})$, we assume that agents whose last observations yield higher value estimates are more informative for coordination. Thus, we determine the generation order $(\kappa(1),\ \kappa(2),\ldots,\kappa(N))$ by sorting agents in descending order of their predicted values at time $t+H-1$, given by a pre-trained value function

$$\text{Order(LOV)} = (\kappa(1),\ \kappa(2),\ldots,\kappa(N)) \ = \ \text{argsort}\big[V_\zeta(o_{t+H-1}^1),\ldots,V_\zeta(o_{t+H-1}^N)\big]_\downarrow, \tag{7}$$

where the subscript $\downarrow$ means "sorted in descending order", and we refer to as the "LOV" (Last Observation Value) ordering. The agent with the highest value is scheduled to generate its trajectory first, and the one with the lowest value last. This order fits well with our subgoal-based generation above. Because the trajectory of the agent with the lowest value is likely to be poor and its value peak is highly likely to be close to the beginning, we have more design freedom for agents with lower values and it is advantageous to schedule trajectory generation for such agents at the later stage of sequential generation.

While LOV focuses on predicted value at last timestep, we consider the second ordering scheme, prioritizing agents based on their "subgoals", where subgoal $g^i$ is defined as the timestep at which agent $i$'s estimated value reaches its peak (defined in Section 3.1). Agents whose value peaks occur earlier in the horizon impose weaker constraints on later behavior. Thus, placing these agents later in the generation order expands the design space and promotes more diverse trajectory synthesis. Consequently, we sort agents in descending order of their subgoal indices ($g^i$):

$$\text{Order(SG)} = (\kappa(1),\kappa(2),\ldots,\kappa(N)) = \text{argsort}\big[g^1,\ldots,g^N\big]_\downarrow, \tag{8}$$

denoted as the "SG" (SubGoal) ordering.

Together, these two ordering strategies allow the sequential generator to adapt to different coordination structures, favoring agents with high value (LOV) or those imposing long-term constraints (SG).

*Remark:* For MASTARS, preserving the relative order of state values is important; otherwise, suboptimal states may be chosen as goals and the generation order will not be set as intended. Considering the fact that we need values only for the observed states in dataset, we use SARSA-style value learning, which relies only on observed transitions $(s, a, r, s', a')$ in the offline dataset, to minimize estimation errors and avoid extrapolation, ensuring stable ordering when rewards are noiseless. See Section 4.5 for this.

In summary, MASTARS provides an end-to-end augmentation framework that generates coherent and high-quality multi-agent trajectories via value-based agent ordering, subgoal-conditioned masking and diffusion-based inpainting. It ranks agents by their estimated final state value, anchors generation after high value subgoals, and fills in

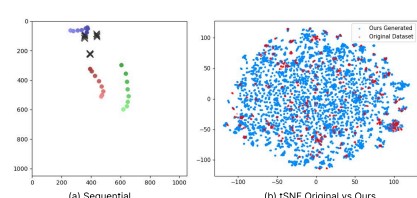

Figure 3: Performance of MASTARS

| Alg | Dataset | Tag | | | Spread | | | World | | | Average |
|---|---|---|---|---|---|---|---|---|---|---|---|
| | | Medium | MD-Replay | Random | Medium | MD-Replay | Random | Medium | MD-Replay | Random | |
| CFCQL | Original | 44.3±16.5 | 43.3±13.7 | 7.4±2.1 | 16.6±9.8 | 31.2±14.0 | 18.8±7.6 | 82.0±25.5 | 49.1±8.5 | 7.9±2.3 | 33.4 |
| | MADiff* | 57.8±16.4 | 54.3±5.8 | 12.2±8.8 | 19.6±10.1 | 58.8±7.7 | 20.7±10.6 | 94.2±9.3 | 54.1±9.4 | 8.6±3.3 | 42.3 |
| | DOF* | 35.0±8.2 | 29.9±4.9 | 9.3±9.2 | 12.5±6.7 | 54.1±14.5 | 15.7±6.6 | 88.8±21.4 | 23.1±8.0 | 3.4±4.8 | 30.2 |
| | MADiTS | 52.8±14.9 | 24.5±6.0 | 6.7±3.0 | 25.8±8.9 | 46.7±12.3 | 21.6±10.5 | 90.8±22.3 | 58.6±5.7 | 3.7±1.2 | 36.8 |
| | INS | 46.6±7.8 | 45.8±8.0 | 8.1±5.4 | 18.5±8.9 | 45.1±14.4 | 25.8±6.7 | 103.8±19.1 | 51.9±10.4 | 8.1±2.8 | 39.3 |
| | Ours-LOV | 58.2±12.0 | 53.4± 6.2 | 14.6±7.8 | 17.6±14.0 | 56.7±14.5 | 21.9±5.1 | 108.5±16.0 | 52.8±13.4 | 10.1±2.2 | 43.7 |
| | Ours-SG | 54.7±17.7 | 54.4±6.0 | 13.3±3.5 | 20.6±9.5 | 59.7±16.6 | 25.2±8.2 | 111.4±10.8 | 51.5±8.0 | 10.0±3.0 | 44.5 |
| OMAR | Original | 10.8±15.1 | 53.3±12.1 | 21.7±11.1 | -1.1±6.4 | 32.8±7.3 | 21.1± 7.0 | 63.5±34.8 | 55.3±8.9 | 7.1±2.3 | 29.4 |
| | MADiff* | 18.4±25.0 | 62.8±8.1 | 17.9±5.3 | -1.0±6.9 | 50.0±10.1 | 18.8±7.6 | 58.0±39.5 | 59.1±18.0 | 11.7±3.4 | 32.9 |
| | DoF* | 7.1±7.7 | 41.1±7.0 | 24.0±7.2 | -2.2±13.7 | 39.4±8.7 | 18.6±8.0 | 68.9±26.6 | 40.9±7.5 | 10.4±2.6 | 27.6 |
| | MADiTS | 23.9±9.5 | 41.5±11.3 | 28.8±10.8 | 3.3±7.6 | 43.7±12.6 | 23.5±11.3 | 53.1±19.3 | 56.1±3.8 | 6.3±2.3 | 31.2 |
| | INS | 4.5±5.6 | 50.9±8.5 | 18.1±5.5 | 0.6±8.9 | 45.4±10.5 | 21.8±5.1 | 75.8±29.3 | 54.4±11.5 | 7.8±3.6 | 31.0 |
| | Ours-LOV | 22.7±19.6 | 66.0±12.9 | 24.6±3.7 | 0.6±7.6 | 46.1±15.0 | 22.3±7.9 | 91.1±8.9 | 63.6±13.6 | 9.8±5.9 | 38.5 |
| | Ours-SG | 24.3±21.7 | 60.9±6.9 | 19.1±7.2 | 9.7±4.8 | 46.5±20.7 | 27.0±10.2 | 109.7±12.4 | 59.5±12.3 | 10.3±5.0 | 40.8 |
| TD3+BC | Original | 23.4±11.3 | 32.2±9.0 | 3.7±3.7 | 8.5±7.3 | 28.1±10.6 | 20.1±7.2 | 84.4±31.3 | 50.6±8.9 | 4.4±2.3 | 28.4 |
| | MADiff* | 37.2±27.3 | 49.2± 9.4 | 13.2±9.2 | 7.5±6.4 | 53.8±18.7 | 15.8±8.1 | 82.5±16.7 | 47.4±10.6 | 7.1±2.5 | 34.9 |
| | DoF* | 22.4±9.8 | 32.8± 7.6 | 3.4±1.3 | 9.6±6.7 | 49.4±18.6 | 19.1±8.9 | 64.5±33.0 | 26.2±8.5 | 9.8±6.0 | 26.4 |
| | MADiTS | 39.9±6.2 | 22.0±4.6 | 2.5±1.2 | 11.0±3.3 | 45.3±10.5 | 17.4±9.0 | 75.0±12.3 | 60.7± 8.8 | 1.5± 2.5 | 30.6 |
| | INS | 15.7±9.7 | 43.9± 7.0 | 4.6±3.7 | 5.6±15.9 | 45.9±20.7 | 21.6±8.2 | 66.5±34.6 | 49.0±12.4 | 8.0±1.8 | 29.0 |
| | Ours-LOV | 41.2±31.0 | 45.0±13.1 | 5.0± 3.3 | 13.8±15.9 | 52.8±13.5 | 21.3±10.0 | 108.8± 9.4 | 61.0±10.6 | 4.4± 1.8 | 39.3 |
| | Ours-SG | 32.8±21.6 | 50.8±8.0 | 10.9±4.6 | 7.8±5.1 | 54.2±17.9 | 24.7±10.1 | 100.2±8.8 | 52.2±10.3 | 8.2±4.1 | 38.0 |

Table 1: Normalized return in MPE environment (bold: best, underline: second best)

missing segments using the RePaint algorithm while preserving good behavior and introducing diverse and coordinated variations. The complete algorithm is provided in Appendix C.1. Fig. 3 shows MASTARS results for the same setup as Fig. 1 in Section 1. Now it is seen that the generated trajectory has the desired coordinated behavior: each of red, green and blue agents follows a distinct mark. Furthermore, it is seen in the t-SNE plot that the generated trajectories show high diversity; the projections of many generated trajectories (blue) do not overlap with those of the original trajectories (red) as compared with Fig.1(d). Augmenting the original dataset with these newly generated improved trajectories can enhance the offline RL performance, as seen in the next section.

## 4 EXPERIMENTS

Our evaluation focuses on six aspects: (i) performance gains with the augmented dataset, (ii) analysis of our generation strategy, (iii) the contribution of key components through ablation,(iv) sensitivity analysis of value estimation, (v) effect of generation order, and (vi) analysis of generated dataset.

### 4.1 EXPERIMENTAL SETUP

We evaluated MASTARS with the two proposed ordering schemes (LOV and SG) using three offline MARL algorithms: CFCQL (Shao et al., 2023), which applies counterfactual conservative regularization; OMAR (Pan et al., 2022), which combines zeroth-order search with first-order gradients; and MA-TD3+BC (Pan et al., 2022), a multi-agent extension of TD3+BC on two standard multi-agent benchmarks: MPE (Lowe et al., 2017a) and SMAC (Samvelyan et al., 2019a). Additional details are in Appendix D, and results on the more challenging SMAC-v2 (Ellis et al., 2023) are reported in Appendix J.

**Baselines.** We compared our method against several data augmentation approaches[2]: 1) Original: using only the offline dataset without augmentation. 2) MADiff* (Zhu et al., 2024): a diffusion-based trajectory generator with inter-agent attention. 3) DoF* (Li et al., 2025): a diffusion-based MARL policy extending the IGM principle to an IGD formulation. 4) MADiTS (Yuan et al., 2025): a stitching-based approach that constructs trajectories by combining segments. 5) INS (Fu et al.): a sparse-attention method that explicitly models inter-agent interactions.

### 4.2 MAIN RESULTS ON OFFLINE MARL BENCHMARKS

As shown in Tables 1 and 2, the two variants of MASTARS (Ours-LOV and Ours-SG) consistently improve performance over the original dataset across all environments and algorithms, whereas all

---

[2]An asterisk (*) indicates methods that are not trajectory generators; we use only their generation component.

| Alg | Dataset | 3m Medium | 3m Good | 2s3z Medium | 2s3z Good | 8m Medium | 8m Good | Average |
|---|---|---|---|---|---|---|---|---|
| CFCQL | Original | 0.75±0.02 | 0.96±0.04 | 0.38±0.38 | 0.40±0.55 | 0.57±0.52 | 0.80±0.45 | 0.64 |
| | MADiff* | 0.78±0.27 | 0.84±0.36 | 0.31±0.18 | 0.68±0.23 | 0.80±0.45 | 0.87±0.13 | 0.71 |
| | DoF* | **0.89±0.14** | 0.86±0.09 | 0.16±0.49 | 0.76±0.33 | 0.78±0.25 | 0.73±0.19 | 0.70 |
| | MADiTS | 0.70±0.14 | 0.86±0.27 | 0.30±0.65 | 0.51±0.14 | 0.90±0.08 | 0.89±0.12 | 0.69 |
| | INS | 0.82±0.31 | 0.79±0.46 | 0.17±0.7 | **0.83±0.21** | 0.82±0.33 | 0.85±0.16 | 0.71 |
| | Ours-LOV | 0.83±0.10 | **0.99±0.01** | **0.45±0.35** | 0.78±0.15 | **1.00±0.00** | **0.97±0.04** | **0.84** |
| | Ours-SG | 0.87±0.13 | 0.97±0.06 | 0.28±0.25 | 0.67±0.12 | 0.84±0.06 | 0.85±0.07 | 0.75 |
| OMAR | Original | 0.71±0.07 | 0.41±0.24 | 0.40±0.55 | 0.60±0.55 | 0.75±0.43 | 0.75±0.43 | 0.60 |
| | MADiff* | 0.71±0.30 | 0.53±0.42 | 0.37±0.11 | 0.53±0.19 | 0.89±0.11 | 0.69±0.38 | 0.62 |
| | DoF* | 0.82±0.18 | **0.91±0.14** | 0.40±0.42 | 0.76±0.19 | 0.87±0.07 | 0.74±0.28 | 0.75 |
| | MADiTS | 0.72±0.12 | 0.44±0.40 | 0.36±0.60 | 0.40±0.29 | 0.72±0.26 | 0.89±0.03 | 0.59 |
| | INS | **0.84±0.2** | 0.88±0.14 | 0.36±0.37 | 0.78±0.35 | 0.85±0.14 | 0.81±0.15 | 0.75 |
| | Ours-LOV | 0.76±0.43 | 0.57±0.33 | **0.55±0.32** | 0.78±0.15 | **1.00±0.00** | **0.99±0.01** | **0.78** |
| | Ours-SG | 0.81±0.08 | 0.79±0.38 | 0.22±0.18 | **0.8±0.45** | 0.96±0.05 | 0.99±0.03 | 0.76 |

Table 2: Test win rate in SMAC environment (bold: best, underline: second best)

other baselines have some combination of algorithm and dataset for which the augmented dataset yields worse performance than the original dataset.

On the MPE benchmark, Ours-LOV improves average performance by 30.8%, 31.0%, and 38.4% for CFCQL, OMAR, and MA-TD3+BC, respectively, relative to the original dataset. Ours-SG shows similar benefits, yielding gains of 33.2%, 38.8%, and 33.8% for the same algorithms.

On SMAC, consistent improvements are also observed under both CFCQL and OMAR, where Ours-LOV achieves the highest or second-highest performance on 9 out of 12 tasks, including nearly a 100% win rate on the 8m map.

Compared with other augmentation baselines, both Ours-SG and Ours-LOV rank either the best or the second-best in nearly all cases, and for the overall average performance across all tasks within each algorithm, Ours-LOV and Ours-SG variants consistently achieve the top two positions. These consistent improvements highlight the superiority of MASTARS in multi-agent offline settings. Implementation details are available in Appendix E, and the computational cost analysis is provided in Appendix F.

## 4.3 VISUALIZATION OF SEQUENTIAL TRAJECTORY GENERATION

We visualize a trajectory generated by MASTARS with LOV ordering scheme to illustrate its sequential construction process. As shown in Fig. 4, each agent's trajectory is generated, following the order determined by the agent's estimated value in the original trajectory: blue, red, and green. Since the blue agent is closest to the target in the raw trajectory, it has the highest value and is generated first. The red agent, being the next closest, follows. This visualization demonstrates that our value-based agent ordering scheme not only functions as intended but also enables the model to progressively generate more coordinated behaviors. As a result, MASTARS produces realistic and cooperative trajectories in which agents successfully avoid collisions and reach their designated targets.

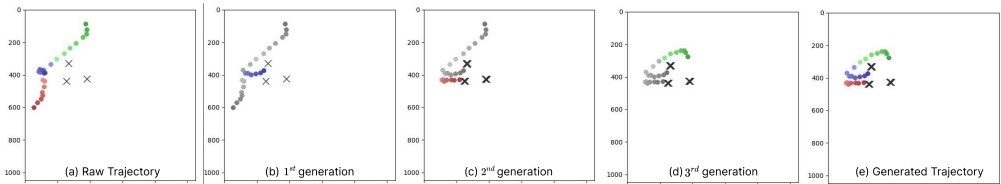

(a) Raw Trajectory    (b) 1st generation    (c) 2nd generation    (d) 3rd generation    (e) Generated Trajectory

Figure 4: Visualization of the full trajectory generation process in our method. Agents are generated sequentially via subgoal-conditioned inpainting. Each panel shows the trajectory state after one agent is generated, starting from the raw trajectory (left) and ending with the fully generated result (right).

## 4.4 ABLATION STUDIES

We conducted ablation studies to assess the contributions of the three key components of MASTARS: (1) subgoal conditioning, (2) sequential generation, and (3) inpainting. Note that all ablation studies

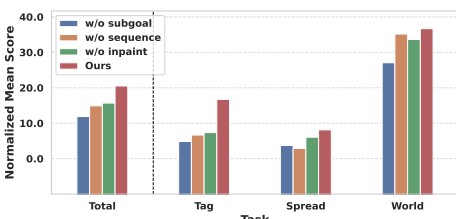

Figure 5: Ablation performance

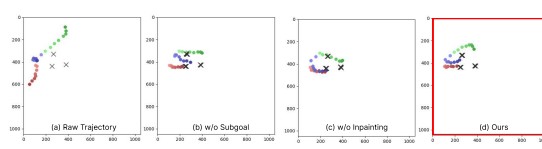

Figure 6: Trajectory comparison under ablation

were performed using the LOV ordering scheme. As shown in Fig. 5, removing any of these modules (denoted as "w/o") leads to a clear drop in performance. Fig. 6 further illustrates their effects: without subgoal conditioning, agents regenerate trajectories from the beginning, unnecessarily replacing optimal segments and producing a suboptimal trajectory, as shown by the green agent; without inpainting, transitions become fragmented and incoherent, as agents fail to align their behaviors with those of others, as illustrated by the collision between the blue and red agents. In contrast, combining all three within an agent-wise sequential generation framework yields stable, goal-directed trajectories aligned with expert behavior. Furthermore, Fig. 1 highlights the importance of sequential generation, showing that joint generation (w/o sequence) without ordering results in poorer coordination. Taken together, these results confirm that each component plays an indispensable role in producing effective and coordinated trajectories for offline MARL.

## 4.5 SENSITIVITY ANALYSIS OF VALUE ESTIMATION

We tested the stability of our SARSA-style value learning. Across MPE benchmarks, normalized TD error steadily converged to near zero (within $\sim 1\%$), as value learning progress, confirming reliable estimation (Table 3). To test sensitivity, we intentionally injected random noise into values during subgoal generation, perturbing each value within $[(1 - \sigma)\text{value}, (1 + \sigma)\text{value}]$. As Table 4 shows, performance dropped with higher noise, underscoring the need for order-preserving estimates. Still, for $\sigma < 0.1$, MASTARS consistently outperformed the original dataset.

Table 3: TD error for value estimation

| Time | 0 | 30K | 60K | 90K |
|---|---|---|---|---|
| Value Loss | 86.3 | 0.02 | 0.01 | 0.01 |

Table 4: Average performance across different noise levels

| Noise Level ($\sigma$) | 0 | 0.05 | 0.1 | 0.15 | Original |
|---|---|---|---|---|---|
| Average Performance | **37.0** | **33.1** | **32.0** | 9.0 | 30.4 |

## 4.6 EFFECT OF GENERATION ORDER.

To evaluate our generation order strategy, we compared (i) Reverse: agents ordered in ascending value, (ii) Random: shuffled order, and (iii) Ours: value-based ordering prioritizing high-value agents, on the MPE Spread task. As shown in Table 5, our method consistently outperforms the alternatives, indicating that generating high-value agents first provides more informative context and yields better coordinated behaviors. The complete results can be found in Appendix K.

Table 5: Average performance with generation orders.

| Order | Reverse | Random | Ours-LO | Ours-SG |
|---|---|---|---|---|
| Performance | 31.55 | 32.07 | **37.04** | 36.13 |

## 4.7 ANALYSIS OF GENERATED DATASET

To evaluate differences from the original data, we adopted the DIRE metric (Wang et al., 2023b), originally proposed in vision domain for distinguishing between original and generated samples. Given an image $x_0$, the metric computes $DIRE(x_0) = |x_0 - R(I(x_0))|$, where $I(\cdot)$ denotes the forward DDIM process (Song et al., 2020b) and $R(\cdot)$ its deterministic reverse. Intuitively, $R(I(x_0))$ is the regenerated version of $x_0$, and DIRE captures the reconstruction error. We trained $I$ and $R$ on the original dataset and applied the metric to both

Table 6: Reconstruct error

| | Original | MADIFF* | Ours |
|---|---|---|---|
| DIRE | 0.045 | 0.086 | **0.102** |

Table 7: Post-subgoal ratio

| | Medium | MD-Replay | Random |
|---|---|---|---|
| spread | 0.58 | 0.26 | 0.62 |
| tag | 0.42 | 0.52 | 0.63 |
| world | 0.44 | 0.17 | 0.35 |

original and generated trajectories in the MPE Spread task. Lower scores indicate closer alignment with the training data, while higher scores reflect greater novelty. As shown in Table 6, MASTARS

trajectories deviate more strongly from the original, which is consistent with the t-SNE visualization in Fig. 3. This demonstrates that MASTARS does not simply clone behaviors, but produces diverse and improved trajectories beyond the original dataset.

In addition, we analyzed how much of each trajectory was newly generated for our partial generation after subgoal time step. Since MASTARS reuses high-quality segments rather than regenerating entire trajectories, the proportion of generated rollout varies. As seen in Table 7, the average generated portion is 44%, indicating that MASTARS strikes a balance between leveraging good segments and synthesizing new ones to produce improved trajectories. More ablation studies including analyses of hyperparameter sensitivity, scalability, and a detailed explanation of the DIRE metric are provided in Appendix G.

## 5 RELATED WORKS

**Offline MARL.** In offline MARL, the growing joint action space creates challenges such as coordination under partial observability, out-of-distribution errors, and convergence to sub-optimal policies. Prior work address these via conservative value estimation (CFCQL (Shao et al., 2023)), hybrid optimization (OMAR (Pan et al., 2022)), and global-to-local regularization (OMIGA (Wang et al., 2023a)). Meanwhile, diffusion models reduce extrapolation errors in single-agent offline RL. Building on this, MADiff (Zhu et al., 2024) extends diffusion to multi-agent settings with attention-based coordination, and DoF (Li et al., 2025) generalizes the Individual-Global-Max (IGM) principle to Individual-Global-Distributed (IGD), aligning multi-agent outcomes with individual distributions.

**Data augmentation.** Data augmentation has been widely explored in vision to improve generalization and robustness, with inpainting methods (e.g., RePaint (Lugmayr et al., 2022), PSM (Li et al., 2023), Context Encoders (Pathak et al., 2016)) leveraging generative models to reconstruct missing regions without retraining. In offline RL, where performance is sensitive to limited data, augmentation strategies include model-based approaches that learn dynamics with uncertainty handling (Kidambi et al., 2020; Yu et al., 2020; 2021; Zhang et al., 2023), and generative methods that synthesize high-quality trajectories (Lu et al., 2023; Li et al., 2024; Lee et al., 2024). In the multi-agent setting, MADiTS (Yuan et al., 2025), EAQ (Oh et al., 2024), and INS (Fu et al.) enhance coordination via trajectory stitching, Q-guided diffusion, and sparse-attention synthesis, respectively. However, as these methods jointly generate all agents' trajectories and struggle to ensure diverse coordinated outcomes, our agent-wise sequential generation enables more flexible and well-coordinated trajectories.

**Sequential Modeling in MARL** Recent work has explored treating MARL as a sequence generation problem, including MAT (Wen et al., 2022), PMAT (Hu et al., 2025), HARL (Zhong et al., 2024), SeqComm (Ding et al., 2024), DIMA (Zhang et al., 2025), and SeqWM (Zhao et al., 2025). These approaches show that imposing an ordering over agents, communications, or latent variables can improve coordination and scalability. However, they target policy learning and message passing rather than trajectory augmentation. In contrast, MASTARS uses a sequential paradigm specifically for generative augmentation and combines it with diffusion-based inpainting to refine trajectories conditioned on other agents, enabling coordinated multi-agent synthesis without joint sampling.

## 6 CONCLUSION

We propose MASTARS, a novel data-augmentation framework for offline MARL that overcomes prior limitations in capturing coordination. It generates trajectories agent-wise and sequentially via diffusion-based inpainting with return-conditioned subgoals and value-based ordering. On MPE and SMAC, MASTARS significantly improves multiple offline MARL algorithms, and qualitative visualizations show more coordinated, realistic behaviors than prior methods.

## ETHICS STATEMENT

This work uses only publicly available benchmark environments (SMAC, MPE) and does not involve human subjects, private data, or sensitive information. The study complies with the ICLR Code of Ethics.

REPRODUCIBILITY STATEMENT

We provide detailed descriptions of models, training, and evaluation in the main paper and appendix. All datasets are public, and anonymized source code with scripts is included in the supplementary materials. Pseudo-code of the proposed method is provided in Appendix C.1.

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

## A  TRAJECTORY GENERATION IN SINGLE-AGENT RL

In offline RL, diffusion models are trained to generate entire trajectory segments by treating each segment as if it were a 2D "image." Concretely, they learn the distribution $p_\theta\big(\tau_{t:t+H-1}\big)$, $\tau = (s_{t:t+H-1}, a_{t:t+H-1})$, by minimizing the DDPM loss in equation 1. To do so, each trajectory segment is reshaped into a matrix of size $(d_s + d_a) \times H$ by stacking state and action vectors as rows, so that the diffusion model generates this matrix just as it would generate an image:

$$\tau = \begin{bmatrix} \mathbf{s}_t & \mathbf{s}_{t+1} & \cdots & \mathbf{s}_{t+H-1} \\ \mathbf{a}_t & \mathbf{a}_{t+1} & \cdots & \mathbf{a}_{t+H-1} \end{bmatrix}. \tag{9}$$

By framing trajectories in this way, the model naturally outputs new trajectory "images" whose statistical properties match those of the offline dataset.

## B  TRAJECTORY GENERATION IN MADIFF

Given a sequence of observations $\tau = (\boldsymbol{o}_{t:t+H-1})$, MADiff (Zhu et al., 2024) proposes two trajectory generation strategies; Independent Trajectory Generation and Joint Trajectory Generation.

**Independent Trajectory Generation**  In the independent scheme, we adopt a decentralized execution manner in MADiff. Each agent $i$ generates its trajectory based solely on its current local observation, without access to the observations or plans of other agents. The diffusion process begins with a noisy trajectory conditioned on agent $i$'s current observation, represented as $\tilde{\boldsymbol{x}}_K = [[\tilde{x}_{K,t}^1, \ldots, \tilde{x}_{K,t_e}^1], \ldots, [o_t^i, \ldots, \tilde{x}_{K,t_e}^i], \ldots, [\tilde{x}_{K,t}^N, \ldots, \tilde{x}_{K,t_e}^N]]$, with $\tilde{x}_{K,t'}^j \sim \mathcal{N}(0,1)$, and $t_e = t+H-1$.

**Joint Trajectory Generation**  In contrast, the joint generation scheme adopts a centralized control setup as in MADiff, where the trajectories of all agents are generated simultaneously, conditioned on the current joint observation $\boldsymbol{o}_t$. The initial noisy trajectory is constructed as $\tilde{\boldsymbol{x}}_K = [[\tilde{o}_{t_s}^1, \ldots, \tilde{x}_{K,t_e}^1], \ldots, [o_{t_s}^i, \ldots, \tilde{x}_{K,t_e}^i], \ldots, [\tilde{o}_{t_s}^N, \ldots, \tilde{x}_{K,t_e}^N]]$, again with $\tilde{x}_{K,t'}^j \sim \mathcal{N}(0,1)$, and $t_e = t_s+H-1$.

## C PSEUDOCODE AND LOSS FUNCTIONS

This section presents the algorithm in detail using pseudocode and outlines the loss functions of the pretrained models used for generation.

### C.1 PSEUDOCODE

---

**Algorithm 1:** Agent-Wise Sequential Trajectory Generation with Return-Guided Subgoal

---

**Input:** Original offline dataset $D_{\text{org}}$, Pretrained diffusion model $\epsilon_\theta$, Inverse dynamics model $I_\phi$, Transition model $f_\varphi$, Reward model $g_\xi$, Value function $V_\zeta$, Number of episodes to generate $N_{\text{episode}}$, Number of denoising steps $K$

**Output:** Augmented dataset $D_{\text{aug}}$

Initialize $D_{\text{aug}} \leftarrow \emptyset$;

**while** *number of episodes in $D_{aug} \leq N_{episode}$* **do**

    Randomly sample joint observation sequence $\tau^0 = (\boldsymbol{o}_t, \dots, \boldsymbol{o}_{t+H+1})$ from $D_{\text{org}}$;

    `// Value-based Ordering`

    Calculate $\text{Order}(\kappa(1), \dots, \kappa(N))$ from $\tau^0$ ;    ▷ `Value-based agent ordering` Eq. 7

    `Input_traj` $\leftarrow \tau^0$;

    `// Agent-wise sequential generation`

    **for** *agent $i$ in $\text{Order}(\kappa(1), \dots, \kappa(N))$* **do**

        Select $g^i$ as the observation in `Input_traj` with the highest value $V_\zeta$.

        Generate mask $m$ with subgoal $g^i$ ;    ▷ `Subgoal mask Eq. 4`

        **for** $k = K$ *to* 1 **do**

            Repaint with Algorithm 2;

        `Input_traj` $\leftarrow \tilde{\tau}_0$ ;    ▷ `Prepare for next agent`

    `Seq_traj` $\leftarrow \tilde{\tau}_0$

    `// Post-preocessing`

    Compute actions and rewards of `Seq_traj` using $I_\phi, g_\xi$;

    Append them to `Seq_traj`;

    **if** *transition validity (Eq. 6) is satisfied* **then**

        Add `Seq_traj` to $D_{\text{aug}}$;

**return** $D_{\text{aug}}$;

---

---

**Algorithm 2:** Repaint

---

**Input:** `Input_traj`, Gaussian model $\epsilon_\theta$, Trajectory from previous denoised step $\tau_k$, Mask $m$

**Output:** $\tilde{\tau}_k$

$\tilde{\tau}_{k-1}^{others} \leftarrow$ Add noise on `Input_traj` ;    ▷ `Forward process with Eq. 5b`

$\tilde{\tau}_{k-1}^{current-agent} \leftarrow$ sampling from $\tilde{\tau}_k$ using $\epsilon_\theta$ ;    ▷ `Reverse process with Eq. 5c`

$\tilde{\tau}_{k-1} \leftarrow$ Masking with $\tilde{\tau}_{k-1}^{others}$ and $\tilde{\tau}_{k-1}^{current-agent}$ ;    ▷ `Repaint with Eq. 5a`

**return** $\tilde{\tau}_{k-1}$

---

## C.2 Loss functions

**Diffusion model.** The model is conditioned on the normalized return $R(\tau)$ to generate observation sequences of length $H$, using the following objective:

$$\mathcal{L}(\theta) = \mathbb{E}_{k,\tau_0 \in \mathcal{D}, \epsilon, R(\tau), \beta} \left[ \left\| \epsilon - \epsilon_\theta \left( \tau_k, k, \ (1 - \beta)R(\tau_0) + \beta \emptyset \right) \right\|^2 \right]. \tag{10}$$

where $\tau := (\boldsymbol{o}_{t:t+H-1})$ is the observation sequence, $\beta \sim \text{Bern}(\lambda)$ is a Bernoulli mask, and $\epsilon \sim \mathcal{N}(0.I)$

**Dynamic models.** To reconstruct full trajectories including actions and rewards, we train three auxiliary models: an inverse dynamics model $I_\phi(o^i, o^{i'})$ for each agent $i$, a reward model $g_\xi(\boldsymbol{o}, \boldsymbol{a})$, and a transition model $f_\varphi(o^i, a^i)$. These models are optimized jointly to ensure consistency with environment dynamics, using the following objective:

$$\mathcal{L}(\xi, \phi) = \mathbb{E}_{\boldsymbol{o}, \boldsymbol{a}, r, \boldsymbol{o}' \sim D} \left[ \left( r - g_\xi(\boldsymbol{o}, \boldsymbol{a}) \right)^2 + \sum_{i=1}^{N} \left( \left( a^i - I_\phi(o^i, o^{i'}) \right)^2 + (o^{i'} - f_\varphi(o^i, a^i))^2 \right) \right] \tag{11}$$

**Value function.** To determine agent generation order and subgoal positions, we train a value function $V_\zeta$ using a temporal-difference (TD) objective:

$$L(\zeta) = \mathbb{E}_{\boldsymbol{o}, \boldsymbol{a}, r, \boldsymbol{o}' \sim D} \left[ \sum_{i=1}^{N} \left( V_\zeta(o^i) - (r + \gamma V_\zeta(o'^i)) \right)^2 \right] \tag{12}$$

# D EXPERIMENTAL DETAILS

## D.1 TASKS

**MPE** *Multi-Agent Particle Environment (MPE (Lowe et al., 2017a))* is a standard benchmark for continuous multi-agent coordination tasks (Lowe et al., 2017b), featuring simple physical dynamics and inter-agent interactions in a 2D space. We evaluated our method on three representative tasks: `Tag`, `Spread`, and `World`, each requiring distinct coordination.

We provide a detailed description of each task below. Fig. 7 illustrates the environments for the tasks described below.

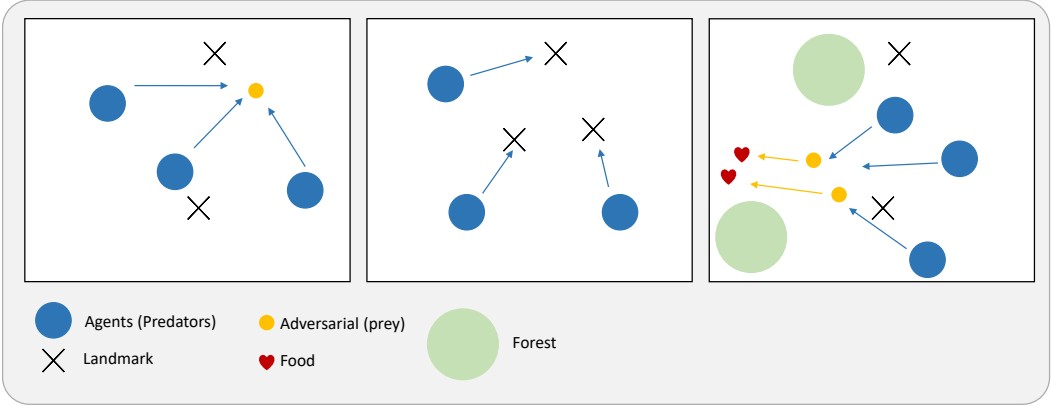

Figure 7: Visualization of MPE task. From left to right: Simple Tag, Simple Spread, Simple World

- `Simple Tag`: This task involves three slower agents (predators) attempting to capture a faster adversarial agent (prey). Due to their speed disadvantage, predators must coordinate their movements to surround and capture the prey, as individual pursuit is ineffective.
- `Simple Spread`: This environment consists of three agents and three fixed landmarks. Agents are rewarded based on their proximity to the landmarks and penalized for collisions with other agents. The objective is to spread out and cover all landmarks while avoiding collisions, requiring spatial coordination.
- `Simple World`: This task involves four cooperative agents and two adversaries. Cooperative agents aim to capture the adversaries, while the adversaries simultaneously try to collect food items scattered throughout the environment and avoid being captured.

**SMAC** *StarCraft Multi-Agent Challenge (SMAC)* (Samvelyan et al., 2019b) is a high-dimensional, discrete-action cooperative benchmark built on the StarCraft II engine. In this benchmark, agents must coordinate to defeat enemies using partial observations and task-specific strategies. We evaluated our method on three commonly used SMAC scenarios: `3m`, `2s3z`, and `8m`.

Table 8: Agent and enemy compositions for SMAC tasks

| Task | Agents | vs | Enemy |
|------|--------|-----|-------|
| 3m | 3 marines | vs | 3 marines |
| 2s3z | 2 stalkers and 3 zealots | vs | 2 stalkers and 3 zealots |
| 8m | 8 marines | vs | 8 marines |

## D.2 Evaluation Algorithms

This section outlines the evaluation algorithms used for comparison on the original and augmented datasets.

**Counterfactual Conservative Q-Learning (CFCQL) (Shao et al., 2023).** CFCQL applies conservative regularization to each agent's Q-function individually in a counterfactual manner—that is, by marginalizing over the actions of other agents while fixing its own. This design enables each agent to estimate a conservative Q-value for its own actions while still accounting for the behaviors of others through the environment dynamics. The training objective is formulated as:

$$\hat{Q}_{k+1} \leftarrow \arg\min_{Q} \; \alpha \left[ \sum_{i=1}^{n} \mathbb{E}_{s \sim \mathcal{D}, \, a^{-i} \sim \beta^{-i}} \mathbb{E}_{a^i \sim \mu^i} \left[ Q(s,a) \right] - \mathbb{E}_{s \sim \mathcal{D}, \, a \sim \beta} \left[ Q(s,a) \right] \right] + \hat{\mathcal{E}}_{\mathcal{D}}(\pi, Q, k) \tag{13}$$

where $\hat{\mathcal{E}}_{\mathcal{D}}(\pi, Q, k)$ denotes the temporal-difference (TD) error used to fit the Q-function.

**Offline Multi-Agent RL with Actor Rectification (OMAR) and MATD3+BC (Pan et al., 2022).** OMAR introduces a hybrid optimization strategy that combines first-order policy gradients with zeroth-order optimization through population-based search. The actor loss is defined as:

$$\mathbb{E}_{\mathcal{D}_i} \left[ (1-\tau) Q_i(o_i, \pi_i(o_i)) - \tau \left\| \pi_i(o_i) - \hat{a}_i \right\|^2 \right] \tag{14}$$

where the candidate actions $\hat{a}_i$ are sampled from a population distribution that is adaptively shaped based on the critic $Q_i$. If the candidate actions $\hat{a}_i$ are instead sampled directly from the offline dataset, the algorithm reduces to MATD3+BC.

## D.3 Training settings.

To effectively investigate the effects of the augmented data, we used a smaller dataset compared to the original offline dataset (Formanek et al., 2023). The dataset size for each task is as follows: each MPE task contains 2,000 episodes with a trajectory $\tau$ length of 25, and each SMAC task contains approximately 250,000 transition samples $(s, a, r, s')$. The diffusion model, value function, and dynamic models were trained using 200K steps and 500K steps, for MPE and SMAC, respectively. All experiments were conducted with five random seeds to ensure statistical reliability with the main results reported as mean $\pm$ standard deviation.

# E IMPLEMENTATION DETAILS AND HYPERPARAMETERS

We present the implementation details and hyperparameter settings that affect the performance and stability of our models during offline training.

**Model Architecture** For the diffusion model, we adopt the same hyperparameter configuration as MADiff (Zhu et al., 2024). All auxiliary networks—including the value function, inverse dynamics model, reward model, and transition model—are implemented as two-layer multilayer perceptrons (MLPs) with ReLU activations and 256 hidden units per layer.

**Hyperparameters for the Post-Processing** This section describes the threshold values used in the post-processing module to filter out invalid transitions based on Eq. 6.

The main results on the MPE environment (Table 1) are obtained using task-specific adaptive thresholds for all baselines, including our method, as specified in Table 9. Since INS does not employ thresholds for trajectory generation, we exclude it from this setting. For the ablation studies, we apply a fixed threshold of 0.03 across all methods to ensure a fair comparison.

For the SMAC and SMAC-v2 benchmark, MADiff*, DoF*, MADiTS, use higher thresholds than MASTARS, as detailed in Table 10, 11. Since they generate entire trajectories, resulting in samples that deviate more significantly from the original dataset distribution. In contrast, our method only generates trajectory segments after the subgoal timestep, enabling the use of lower thresholds and stricter acceptance criteria.

| Task \ Algo | MADiff* | DoF* | MADiTS | Ours-LOV | Ours-SG |
|---|---|---|---|---|---|
| Tag-Medium | 0.03 | 0.02 | 0.03 | 0.02 | 0.008 |
| Tag-MD-Replay | 0.03 | 0.02 | 0.03 | 0.04 | 0.006 |
| Tag-Random | 0.04 | 0.03 | 0.03 | 0.01 | 0.007 |
| Spread-Medium | 0.02 | 0.02 | 0.03 | 0.004 | 0.01 |
| Spread-MD-Replay | 0.02 | 0.02 | 0.03 | 0.0005 | 0.004 |
| Spread-Random | 0.03 | 0.02 | 0.03 | 0.005 | 0.001 |
| World-Medium | 0.04 | 0.04 | 0.01 | 0.02 | 0.001 |
| World-MD-Replay | 0.02 | 0.04 | 0.03 | 0.0005 | 0.003 |
| World-Random | 0.04 | 0.04 | 0.03 | 0.006 | 0.009 |

Table 9: MPE thresholds in all algorithms

| Task \ Algo | MADiff* | DoF* | MADiTS | Ours-LOV | Ours-SG |
|---|---|---|---|---|---|
| 3m-Medium | 0.3 | 0.05 | 1.0 | 0.001 | 0.001 |
| 3m-Good | 0.3 | 0.1 | 1.0 | 0.001 | 0.001 |
| 2s3z-Medium | 0.7 | 0.3 | 2.0 | 0.001 | 0.0001 |
| 2s3z-Good | 0.7 | 0.4 | 2.0 | 0.001 | 0.0001 |
| 8m-Medium | 1.0 | 0.4 | 2.0 | 0.001 | 0.001 |
| 8m-Good | 1.0 | 0.8 | 2.0 | 0.0001 | 0.0001 |

Table 10: SMAC thresholds in all algorithms

**Hyperparameters on Sequential Inpainting** Across all experiments, including both MPE and SMAC, we generate 1,000 augmented trajectories, each consisting of 8 timesteps (i.e., $H = 8$). The diffusion model performs 200 denoising steps during sampling for both MADiff* and MASTARS. For MADiTS, we follow the original hyperparameter settings from the paper for training and sampling, except for the adaptive threshold and the number of generated trajectories, which are adjusted to ensure a fair comparison.

| Task \ Algo | MADiff* | DoF* | MADiTS | Ours |
|---|---|---|---|---|
| Protoss 3_vs_3-Medium | 1.0 | 40 | 5.0 | 0.2 |
| Protoss 3_vs_3-Good | 1.0 | 40 | 5.0 | 0.7 |
| Protoss 5_vs_5-Medium | 1.0 | 40 | 5.0 | 0.1 |
| Protoss 5_vs_5-Good | 1.0 | 40 | 5.0 | 1.0 |
| Terran 3_vs_3-Medium | 1.0 | 100 | 1.0 | 0.4 |
| Terran 3_vs_3-Good | 1.0 | 100 | 1.0 | 0.7 |
| Terran 5_vs_5-Medium | 1.0 | 100 | 5.0 | 0.4 |
| Terran 5_vs_5-Good | 1.0 | 100 | 5.0 | 0.01 |
| Zerg 3_vs_3-Medium | 1.0 | 150 | 5.0 | 0.7 |
| Zerg 3_vs_3-Good | 1.0 | 150 | 1.0 | 0.7 |

Table 11: SMAC-v2 thresholds in all algorithms

## F COMPUTATIONAL COST

All experiments were conducted on a single machine with 8 NVIDIA GeForce RTX 3090 GPUs, 96 logical CPU cores, and Linux (kernel 5.15), using Python 3.8. Training was implemented in Python with CUDA acceleration. To evaluate computational efficiency, we compared the runtime of MASTARS against the MADiTS baseline (based on MADiff) on the MPE Simple Spread (MDRep) task. Both methods were trained for 100K environment timesteps. MADiTS required approximately 3 hours and 7 minutes (186 minutes), while MASTARS completed training in 2 hours and 39 minutes (159 minutes), achieving a 15% reduction in training time under identical conditions.

# G    ADDITIONAL EXPERIMENTAL ANALYSES

## G.1    EFFECT OF HYPER-PARAMETERS

We analyze the sensitivity of MASTARS to two key hyperparameters: the acceptance threshold $\epsilon$ and the number of generated trajectories, both of which influence the diversity of augmented data. As shown in Fig. 8, MASTARS is robust to variations in $\epsilon$, while the number of generated trajectories has a more significant impact. Generating too few samples yields limited improvement, whereas excessive augmentation can introduce distributional shift and degrade performance. Optimal results are achieved with a moderate amount of augmentation, highlighting the importance of balancing quantity and quality. Notably, MASTARS consistently outperforms the original dataset across all settings.

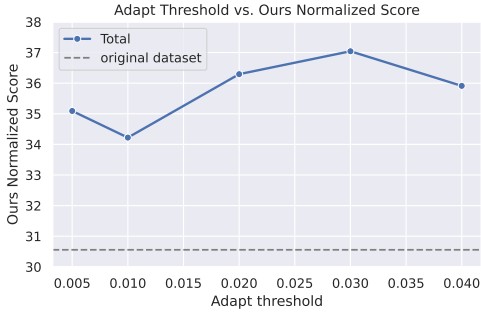 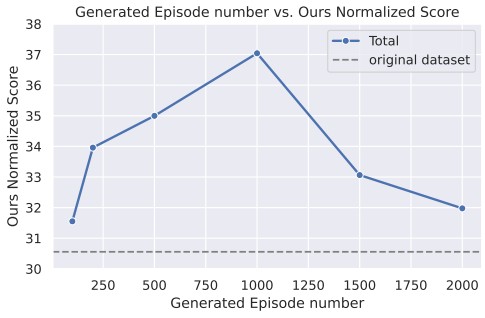

(a) Performance across acceptance threshold $\epsilon$.

(b) Performance with different amounts of generated trajectories.

Figure 8: Analysis of generation sensitivity. (a) shows that our method is robust to threshold selection. (b) shows that moderate augmentation leads to better performance.

## G.2    SCALABILITY ANALYSIS W.R.T. AGENT NUMBER AND HORIZON LENGTH

To verify scalability in terms of agent number and horizon length, we conducted additional experiments in SMAC: the 25m scenario (25 agents) for agent scalability, and the 2c_vs_64zg scenario (horizon length 400) for temporal scalability. As shown in the table 12, MADiTS and MADiFF showed performance drops in these challenging scenarios, while MASTARS consistently outperformed the original offline datasets. These results highlight the effective scalability of MASTARS, demonstrating strong performance even in scenarios with many agents and extended horizons.

Table 12: Test win rate on 25m and 2c_vs_64zg in SMAC environment

| Task \Algorithms | Original | MADIFF* | MADiTS | MASTARS (Ours) |
|---|---|---|---|---|
| 25m | 0.7 | 0.75 | 0.6 | **0.85** |
| 2c_vs_64zg | 0.52 | 0.7 | 0.05 | **0.83** |

## G.3    MASTARS'S ADVANTAGE IN DATA GENERATION

A key strength of diffusion models is that, although they are trained to fit the data distribution, they can still generate slightly diverse, high-quality trajectories by sampling from noise. While MASTARS leverages a diffusion model that aligns with the dataset, our proposed sampling method enables the generation of trajectories that are more diverse and coordinated than return-guided diffusion models. Specifically, if the dataset is suboptimal, a standard return-guided diffusion model jointly generates trajectories that conform closely to this suboptimal distribution, yielding similarly suboptimal outcomes. In contrast, MASTARS sequentially generates trajectories agent-by-agent via inpainting, explicitly conditioning each agent's behavior on previously generated trajectories. This approach allows MASTARS to break free from suboptimal constraints, effectively guiding the generation toward more optimal trajectories.

To assess the difference from the original data, we adopted the DIRE (Wang et al., 2023b) method, which introduces the DIRE metric to distinguish between real and generated images. For a given image $x_0$, DIRE is measured by the following equation:

$DIRE(x_0) = |x_0 - R(I(x_0))|$ where $I(\cdot)$ is the forward DDIM (Song et al., 2020a) process, and $R(\cdot)$ is the deterministic reverse process. Thus, $R(I(x_0))$ is the regenerated image from the encoding of $x_0$. We applied the DIRE metric to both generated and original trajectories in the MPE Spread environment to measure reconstruction error. We trained $I(\cdot)$ and $R(\cdot)$ with original dataset. Lower values indicate closer alignment with the original data, while higher values reflect greater novelty from original data. As shown, indeed, the dataset generated by MASTARS deviates from the original dataset.

Table 13: Reconstruct error

|  | Original Dataset | MADIFF* | MASTARS (Ours) |
|---|---|---|---|
| Reconstruct error | 0.045 | 0.086 | **0.102** |

## H    Limiations and Broader Impact

**Limitations and Future Works**    First, the sequential generation process in MASTARS scales with the number of agents, which may raise concerns about sampling efficiency. However, since the computational cost of sampling is negligible compared to training, this overhead is not a practical limitation. In addition, it would be beneficial to explore alternative approaches for subgoal generation, such as learning subgoal distributions or leveraging a combination of joint and individual value functions (Jeon et al., 2022). Second, MASTARS relies on pre-trained value functions for subgoal selection and agent ordering, which may introduce bias when the dataset quality is low. This issue could be mitigated by iteratively retraining the value function using high-quality generated trajectories. Third, the absence of stitching techniques limits the model's ability to explore behaviors beyond the original data distribution. To enable controlled exploration while staying close to the dataset, MASTARS could be combined with stitching-based augmentation methods.

**Broader Impact**    Our algorithm can generate cooperative trajectories from small datasets, reducing the need for expensive data collection in real-world applications. This has significant potential in domains where reinforcement learning is applicable but data acquisition is costly, such as autonomous driving, warehouse robotics, and surgical training simulators. By lowering the data requirements, our method enables faster prototyping and broader accessibility of RL-based systems. However, careful validation is necessary to ensure that the generated data does not introduce unrealistic behaviors or biases when deployed in safety-critical environments.

## I    Visualization of Generated Trajectories

We provide additional visualization examples across various tasks(Spread and Tag). Each row corresponds to one of eight rollout samples. The leftmost column shows the original trajectory from the dataset, while the three middle columns illustrate intermediate steps of agent-wise sequential generation. The rightmost column (highlighted in red) presents the final trajectory obtained after all agents are sequentially generated using subgoal-conditioned inpainting. These examples demonstrate that MASTARS produces cooperative and diverse trajectories that remain close to the original data distribution, contributing to improved performance through effective augmentation.

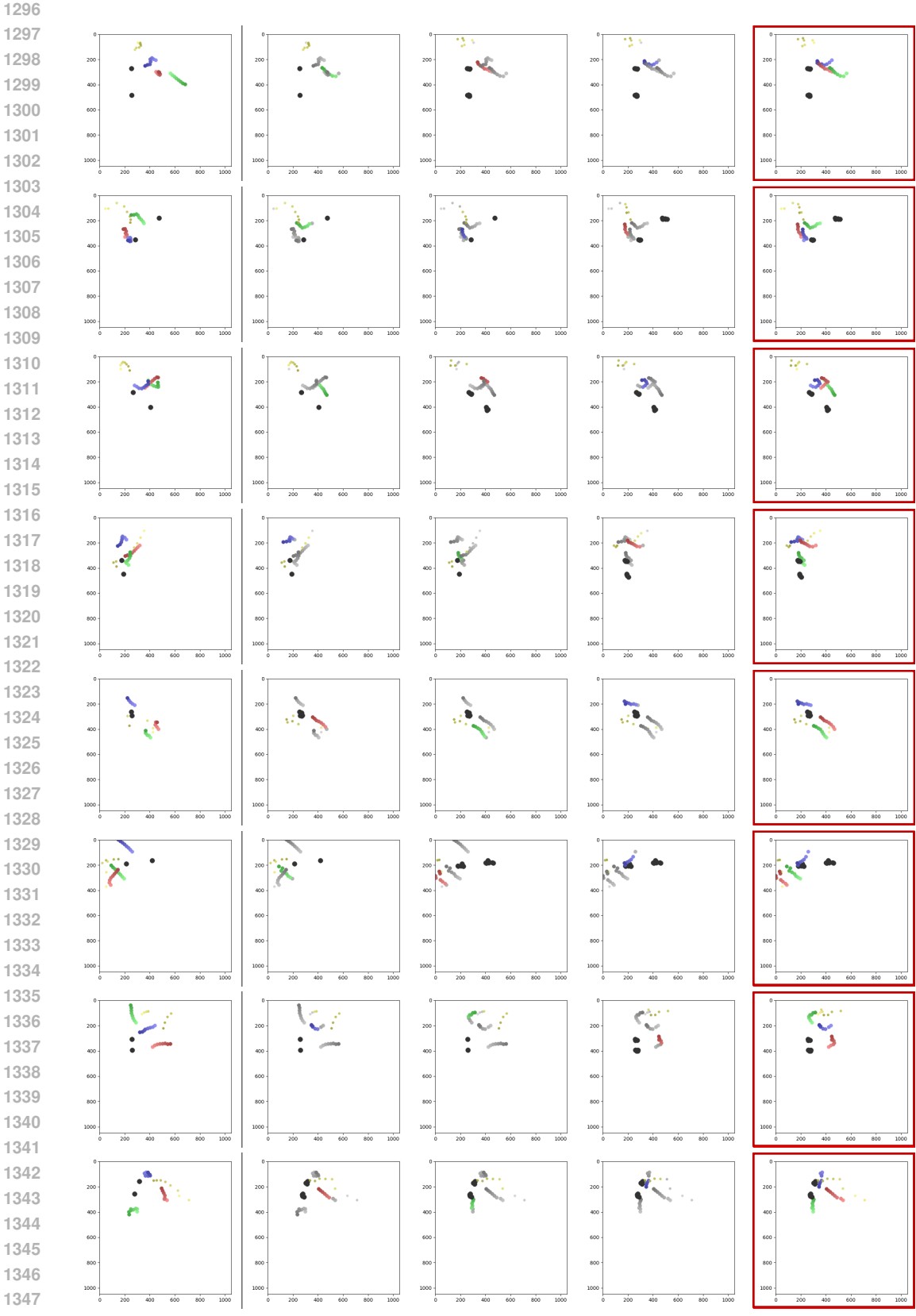

Figure 9: Visualization of the full trajectory generation process using our MASTARS framework in the `Tag-MDRep` task.

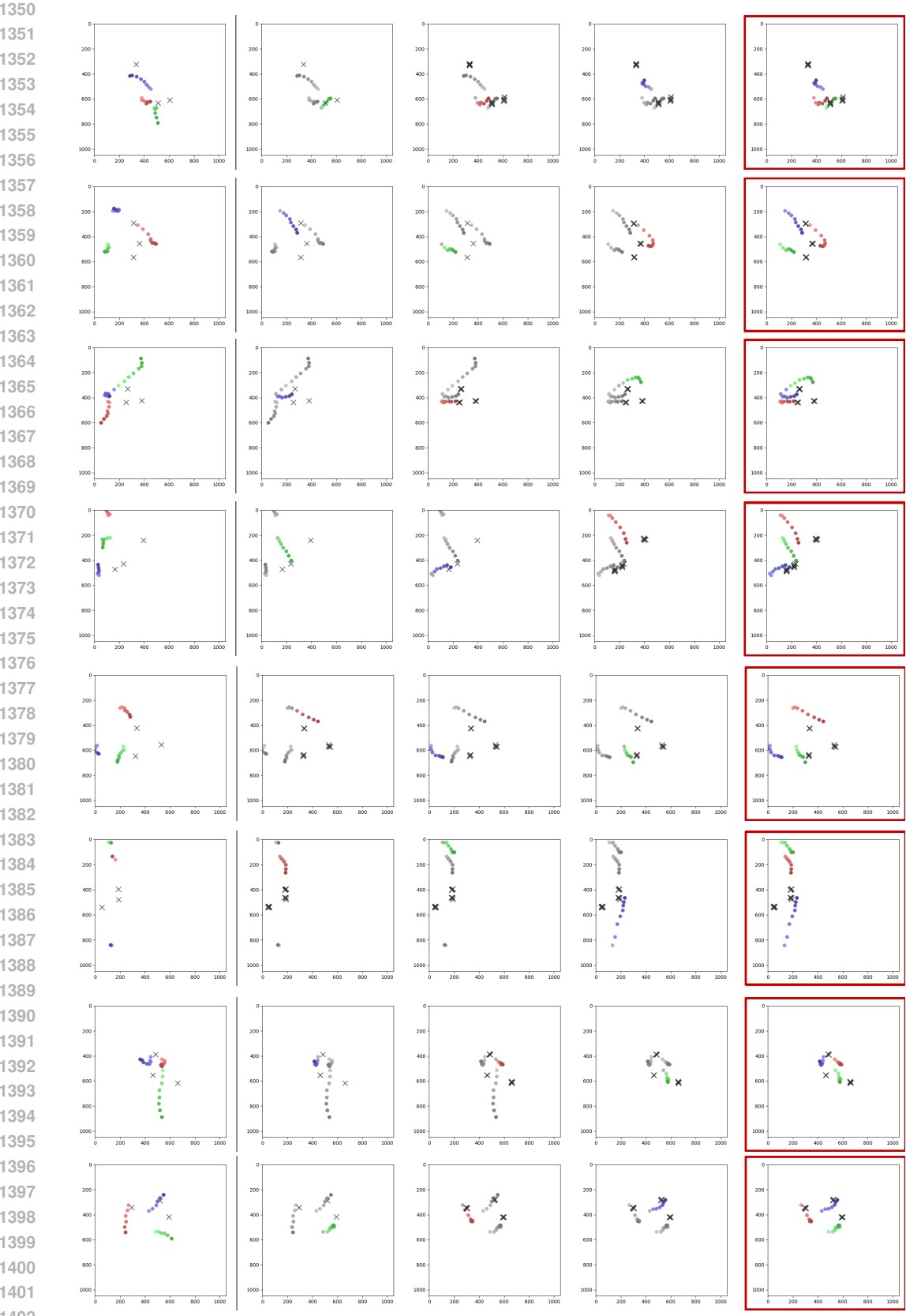

Figure 10: Visualization of the full trajectory generation process using our MASTARS framework in the Spread-MDRep task.

## J    SMAC-V2

This appendix provides additional implementation details and extended experimental results that complement the main paper. In particular, we report results on the more challenging SMAC-v2 (Ellis et al., 2023) benchmark to further evaluate the generalizability of MASTARS beyond the settings considered in the main experiments.

Compared to SMAC, which uses fixed unit types and deterministic starting configurations, SMAC-v2 introduces substantially greater variability: both ally and enemy unit compositions are sampled from probability distributions, and initial agent placements are randomized through reflections and spatial perturbations of the map layout. These features make SMAC-v2 more stochastic and diverse across episodes, providing a stronger testbed for assessing robustness and generalization in multi-agent coordination.

Table 14 summarizes the unit-type sampling distributions used to generate our datasets for each faction (Protoss, Terran, and Zerg). These stochastic unit compositions, combined with randomized starting positions, provide diverse combat scenarios that challenge the robustness and generalization ability of multi-agent reinforcement learning methods.

| Faction | Unit Type | Probability | Start Position (p=0.5) |
|---------|-----------|-------------|------------------------|
| Protoss | Stalker | 0.45 | |
| | Zealot | 0.45 | Surrounded / Reflected |
| | Colossus | 0.10 | |
| Terran | Marine | 0.45 | |
| | Marauder | 0.45 | Surrounded / Reflected |
| | Medivac | 0.10 | |
| Zerg | Zergling | 0.45 | |
| | Baneling | 0.45 | Surrounded / Reflected |
| | Hydralisk | 0.10 | |

Table 14: Unit-type sampling probabilities and starting-position distributions for SMAC-v2 datasets.

To construct the SMAC-v2 datasets, we collected 2,000 episodes for each task, with all episodes having a fixed horizon of 200 steps. To obtain these trajectories, we used the QMIX algorithm (Rashid et al., 2020) implemented in the PyMARL framework, which is a widely adopted baseline in multi-agent reinforcement learning.

Due to the increased stochasticity and difficulty of SMAC-v2, the collected datasets do not achieve near-expert performance (i.e., a test win rate close to 1.0). Instead, we provide two dataset qualities, medium and good, reflecting different levels of policy competence under these challenging settings. Detailed statistics for each task, including the number of trajectories, average return, and test win rate, are reported in Table 15.

Table 16 summarizes the performance of all methods on the SMAC-v2 tasks. MASTARS consistently outperforms existing baselines across both CFCQL and OMAR. Notably, the synthetic data generated by MASTARS leads to substantial performance improvements, achieving a 96% increase under CFCQL and a 114% increase under OMAR.

The relative gains over other generative baselines are also significant. Under CFCQL, MASTARS achieves improvements of 71.2%, 839.5%, 2425%, and 20.6% over MADiFF*, DoF*, MADiTS, and INS, respectively. A similar trend appears under OMAR, with increases of 90.9%, 328.3%, 1092.9%, and 128.8% over the same baselines. When aggregating results across both algorithms, MASTARS maintains strong gains, outperforming MADiFF*, DoF*, MADiTS, and INS by 79.6, 509.9, 1577.3, and 53.4%, respectively.

Table 15: Dataset statistics for SMAC tasks.

| Task | Quality | # Trajectories | Average return | Average win rate |
|---|---|---|---|---|
| protoss 3_vs_3 | medium | 2000 | 16.44 | 0.42 |
| | good | 2000 | 18.16 | 0.58 |
| protoss 5_vs_5 | medium | 2000 | 17.98 | 0.41 |
| | good | 2000 | 19.38 | 0.64 |
| terran 3_vs_3 | medium | 2000 | 11.6 | 0.44 |
| | good | 2000 | 16.2 | 0.63 |
| terran 5_vs_5 | medium | 2000 | 13.09 | 0.4 |
| | good | 2000 | 12.26 | 0.8 |
| zerg 3_vs_3 | medium | 2000 | 9.75 | 0.43 |
| | good | 2000 | 13.33 | 0.57 |

Overall, MASTARS ranks within the top two methods on 9 out of 10 tasks under CFCQL, and achieves a top-two ranking on all tasks under OMAR, reflecting its robustness across diverse scenarios and evaluation settings.

Table 16: Performance on SMACv2 tasks (bold: best, underline: second best). Mean ± std.

| Alg | Dataset | Protoss 3_vs_3 | | Protoss 5_vs_5 | | Terran 3_vs_3 | | Terran 5_vs_5 | | Zerg 3_vs_3 | | Average |
|---|---|---|---|---|---|---|---|---|---|---|---|---|
| | | Medium | Goood | Medium | Goood | Medium | Goood | Medium | Goood | Medium | Goood | |
| CFCQL | Original | 0±0 | 0.31±0.05 | 0.14±0.07 | 0.18±0.07 | 0.23±0.16 | 0.40±0.55 | 0.15±0.25 | 0.15±0.21 | 0.20±0.45 | 0.30±0.17 | 0.21 |
| | MADiFF* | 0.26±0.16 | 0.27±0.15 | 0.06±0.09 | 0.10±0.24 | 0.39±0.34 | 0.56±0.44 | 0.23±0.04 | 0.14±0.19 | 0.10±0.10 | 0.25±0.18 | 0.24 |
| | DoF* | 0.07±0.07 | 0.02±0.02 | 0.01±0.04 | 0.04±0.04 | 0.01±0.01 | 0±0 | 0.05±0.05 | 0.21±0.44 | 0±0 | 0.02±0.02 | 0.04 |
| | MADiTS | 0.07±0.22 | 0.05±0.07 | 0.02±0.03 | 0±0 | 0.06±0.09 | 0.05±0.07 | 0.02±0.02 | 0.0±0.0 | 0.02±0.04 | 0.02±0.03 | 0.03 |
| | INS | 0.16±0.13 | **0.40±0.52** | **0.42±0.52** | 0.20±0.45 | 0.44±0.25 | 0.45±0.36 | 0.40±0.55 | 0.45±0.51 | 0.23±0.09 | 0.20±0.45 | 0.34 |
| | Ours | **0.40±0.55** | 0.19±0.18 | 0.20±0.23 | **0.63±0.51** | 0.48±0.49 | **0.68±0.36** | 0.49±0.48 | 0.42±0.36 | 0.23±0.22 | **0.32±0.06** | **0.4** |
| OMAR | Original | 0±0 | 0.13±0.07 | 0.10±0.06 | 0.06±0.04 | 0.12±0.07 | 0.28±0.22 | **0.31±0.40** | 0.13±0.12 | 0.22±0.43 | **0.21±0.19** | 0.16 |
| | MADiFF* | 0.15±0.08 | 0.16±0.13 | 0.21±0.44 | 0.01±0.01 | **0.48±0.48** | 0.37±0.36 | 0.11±0.07 | 0.08±0.08 | 0.06±0.06 | 0.12±0.09 | 0.18 |
| | DoF* | 0.02±0.04 | **0.44±0.47** | 0.01±0.02 | 0.01±0.03 | 0.10±0.06 | 0±0 | 0.06±0.07 | 0.03±0.04 | 0.04±0.06 | 0.07±0.05 | 0.08 |
| | MADiTS | 0.06±0.06 | 0.02±0.03 | 0.03±0.05 | 0.03±0.04 | 0.05±0.07 | 0.05±0.06 | 0.0±0.0 | 0.02±0.03 | 0.06±0.08 | 0.03±0.04 | 0.03 |
| | INS | 0.08±0.07 | 0.20±0.45 | 0.20±0.45 | 0±0 | 0.21±0.04 | 0.18±0.05 | 0±0 | 0.45±0.48 | 0.14±0.23 | 0±0 | 0.12 |
| | Ours | **0.40±0.55** | **0.44±0.52** | 0.20±0.10 | **0.41±0.54** | **0.48±0.50** | 0.35±0.38 | 0.16±0.09 | **0.46±0.41** | **0.28±0.41** | 0.16±0.12 | **0.33** |

## K  COMPLETE EXPERIMENTAL RESULTS FOR GENERATION ORDER STRATEGIES

We provide the complete experimental results for the generation order strategies, namely reverse (agents ordered in ascending value), random (shuffled order), ours-LOV (value-based ordering), and ours-SG (subgoal-based ordering). As shown in Table 17, our approach consistently outperforms the alternatives, demonstrating that our generation order strategies provide more informative context and lead to better coordinated behaviors.

| Alg | Ordering | Tag | | | Spread | | | World | | | Average |
|---|---|---|---|---|---|---|---|---|---|---|---|
| | | Medium | MD-Replay | Random | Medium | MD-Replay | Random | Medium | MD-Replay | Random | |
| CFCQL | Reverse | 40.3±13.2 | 28.7±6.2 | 9.9±3.9 | **19.1±11.3** | 46.4±12.2 | **23.2±4.7** | 86.8±18.8 | 44.0±6.8 | 7.8±2.2 | 34.0 |
| | Random | 52.1±8.1 | 35.4±10.8 | **13.6±6.6** | 15.6±9.4 | 55.6±18.8 | 22.0±8.2 | 90.2±5.6 | 38.7±7.5 | 8.8±10.7 | 36.9 |
| | Ours-LOV | **50.7±11.0** | **55.5±14.6** | 13.0±5.7 | 12.0±8.4 | 52.8±17.7 | 20.8±7.6 | **95.4±28.1** | 48.9±6.6 | **9.7±3.1** | **39.9** |
| | Ours-SG | 46.7±9.3 | 46.7±14.4 | 10.9±5.5 | 18.0±8.4 | **58.1±13.1** | 18.2±4.7 | 92.9±39.2 | **51.6±8.0** | 9.6±4.4 | 39.2 |
| OMAR | Reverse | 12.1±3.2 | 40.2±12.6 | 18.9±11.3 | -3.8±11.1 | 46.5±16.1 | **25.5±10.0** | 65.8±32.8 | 45.5±3.9 | 9.3±5.7 | 28.9 |
| | Random | **25.4±13.8** | 40.5±7.2 | 16.4±7.6 | -3.5±3.7 | 43.6±10.0 | 24.7±12.2 | 68.4±11.2 | 38.5±16.1 | **11.6±1.6** | 29.5 |
| | Ours-LOV | 15.6±4.7 | 57.4±17.3 | 19.5±4.3 | -2.2±7.4 | **48.4±19.9** | 18.2±10.5 | **93.7±27.2** | 47.0±6.5 | 11.4±5.2 | 34.3 |
| | Ours-SG | 20.0±14.5 | **57.5±6.0** | **20.2±7.2** | **4.7±8.1** | 45.0±14.3 | 19.4±10.6 | 88.6±25.1 | **59.4±12.3** | 9.0±3.7 | **36.0** |
| TD3+BC | Reverse | 25.4±6.8 | 34.2±9.3 | **10.4±4.0** | 2.1±11.6 | **54.0±13.5** | 22.1±7.3 | 79.4±23.4 | 49.5±4.2 | **8.7±1.4** | 31.7 |
| | Random | **32.7±13.8** | 35.5±33.3 | 8.1±6.1 | -5.0±2.2 | 50.5±19.8 | 22.9±6.1 | 76.5±21.0 | 39.6±11.6 | 7.8±1.9 | 29.8 |
| | Ours-LO | 28.8±7.5 | **42.5±14.1** | 5.9±2.6 | **5.6±9.6** | 53.6±15.6 | 20.0±7.4 | **116.1±25.5** | **57.2±9.7** | 2.8±2.6 | **36.9** |
| | Ours-SG | 23.9±8.6 | 41.4±15.0 | 8.3±3.6 | 4.0±2.6 | 50.1±14.5 | **24.5±6.6** | 87.8±35.8 | 52.3±10.3 | 6.6±3.6 | 33.2 |

Table 17: Normalized return in MPE environment (bold: best) with various generation order strategies

## L   VISUAL COMPARISON BETWEEN MADiTS AND OUR METHOD

To further analyze the differences between our method and other joint generation approaches, we provide a visual comparison with MADiTS. Although the main paper includes comparisons against MADiFF*, it is important to clarify that MADiFF is not a trajectory generation method. To the best of our knowledge, among diffusion-based approaches applicable to multi-agent offline reinforcement learning, only two methods are designed to synthesize new data: MADiTS and INS. We therefore compare to MADiTS, which is the only prior work that performs genuine joint trajectory generation. We do not compare against INS, since INS produces transitions rather than trajectories.

As shown in Figure 11, MADiTS frequently generates suboptimal trajectories. In batch 0, the red and green agents move toward the same destination, which reflects a suboptimal cooperative policy. A similar pattern is observed in batch 1 where the blue and red agents collapse to the same target. In contrast, our method consistently guides agents toward distinct destinations in both batches, aligning with the optimal coordinated strategy.

Furthermore, MADiTS often fails to generate valid destination points (marked with X in the figure) and in some cases even fails to produce a feasible trajectory. For example, in Batch 2, the trajectory generated for the blue agent becomes infeasible, whereas our method consistently produces coherent and well-separated target positions. This discrepancy stems from the fundamental differences in generation mechanisms. MADiTS relies on joint conditional diffusion to produce an entire trajectory in a single pass; as a result, its one-shot refinement process often lacks consistency, making the generated destinations noisy and poorly aligned with the surrounding agent behaviors.

In contrast, MASTARS adopts a RePaint-style inpainting strategy. By iteratively resampling masked segments conditioned on neighboring agents and previous denoising steps, our approach progressively corrects local inconsistencies and produces coherent, well-structured trajectories. This iterative refinement allows MASTARS to generate reliable and harmonized trajectory that MADiTS frequently fails to achieve.

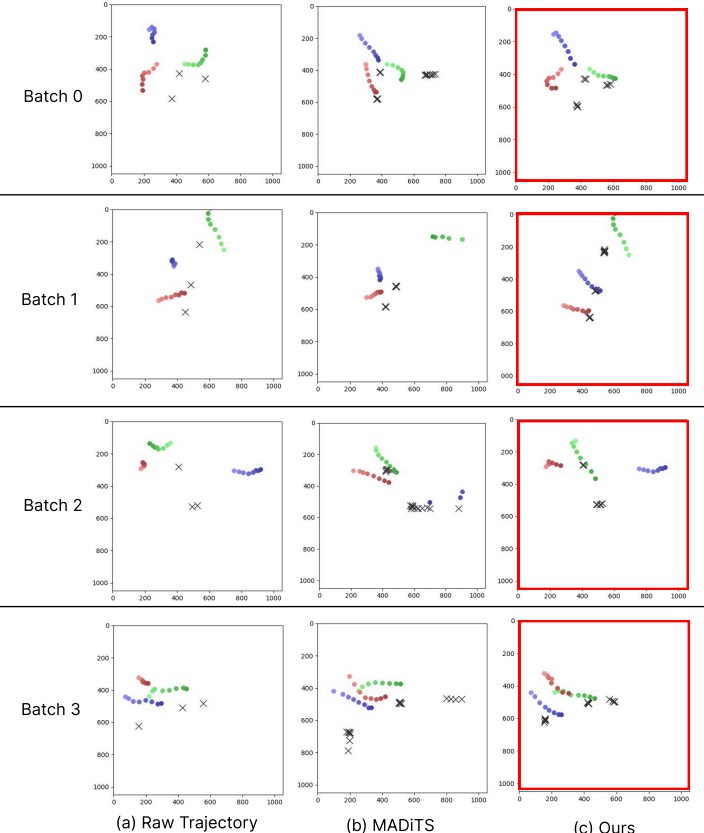

Figure 11: (a) Raw trajectory, (b) MADiTS, and (c) Ours

## M   THE USE OF LARGE LANGUAGE MODELS (LLMs)

Large language models (LLMs) were used as assistive tools for polishing writing, improving clarity, and checking grammar. LLMs did not contribute to research ideation, experimental design, or analysis, and were not involved in generating any research results.

