# OpenReview forum: "MASTARS: Multi-Agent Sequential Trajectory Augmentation with Return-Conditioned Subgoals"
_ICLR.cc/2026/Conference — Submitted to ICLR 2026_

### Official Review · Reviewer_Q8n9 · 2025-10-27

**Soundness:** 3
**Presentation:** 3
**Contribution:** 2
**Rating:** 4
**Confidence:** 3

**Summary:**

This paper proposes a diffusion-based inpainting mechanism for data augmentation in multi-agent offline reinforcement learning. The method aims to enhance both the quantity and quality of offline data to facilitate more effective policy learning.

**Strengths:**

1. The paper is well-organized and logically structured, making it easy to follow.

2. The experimental results are promising.

**Weaknesses:**

1. As the authors mention, there have already been several studies on data augmentation in multi-agent reinforcement learning. However, the Introduction mainly highlights the advantages of MADiff itself, without clearly articulating what specific advantages it has compared to prior works.

2. Although the authors provide extensive experimental validation of the method’s effectiveness, the evaluation is not conducted in more challenging environments such as SMAC2 or settings involving a larger number of agents.

3. The Method section introduces three key components, yet the ablation study does not include an analysis of the third component—Value-based Agent Ordering.

**Questions:**

1. What are the main differences and advantages of this work compared to other data augmentation approaches in multi-agent reinforcement learning?

2. Could the authors provide results on more challenging scenarios to better validate the robustness of the proposed method?

3. Why is there no ablation study for Value-based Agent Ordering in Figure 5?

---

> ### Author Response · Authors · 2025-11-24
>
> We thank the reviewer for the insightful comments and hope that our response together with the revised paper satisfies all concerns of the reviewer. The numbers of equations, figures, tables and sections below refer to those in the revised paper available at openreview unless we give special remarks. Our response to each comment are presented below.
>
> &nbsp;
>
> **R1. Advantage compared to prior works (weakness 1, question 1)**
>
> To the best of our knowledge, among diffusion-based methods applicable to offline MARL data augmentation, only MADiTS[1] and INS[2] explicitly aim to expand logged datasets. MADiff[3] and DoF[4], although diffusion-based, were originally designed to synthesize high-return trajectories and infer actions via inverse dynamics rather than to augment offline data; thus, we apply only their trajectory generators for a fair comparison. Importantly, all four prior approaches (MADiff, DoF, INS, and MADiTS) generate all agents jointly, which requires exploring a trajectory space of size $O(|S|^N)$. This joint sampling quickly becomes difficult to optimize when the offline dataset is limited or suboptimal, and it struggles to scale as the number of agents increases.
>
> In contrast, MASTARS removes joint sampling entirely. By generating agents sequentially and conditioning each one on previously generated agents, MASTARS reduces the search space from $O(|S|^N)$ to $O(|SN|)$ while still modeling inter-agent dependencies. Moreover, the RePaint-style inpainting mechanism iteratively resamples masked segments conditioned on surrounding agents and past denoising steps, allowing coordinated and harmonized behavior to emerge during the generation process—rather than relying on a one-shot refinement of selected agent dimensions.
>
> Taken together, prior works use diffusion models to refine or improve trajectories after joint generation, whereas MASTARS replaces joint generation altogether with a scalable generation framework that supports coordination throughout sampling. This constitutes a qualitatively different methodological direction rather than an incremental extension of existing approaches.
>
> &nbsp;
>
> [1] Yuan et al. Efficient Multi-agent Offline Coordination via Diffusion-based Trajectory Stitching. ICLR 2025.
>
> [2] Fu, Yuqian, et al. "INS: Interaction-aware Synthesis to Enhance Offline Multi-agent Reinforcement Learning." The Thirteenth International Conference on Learning Representations.
>
> [3] Zhu, Zhengbang, et al. "Madiff: Offline multi-agent learning with diffusion models." Advances in Neural Information Processing Systems 37 (2024): 4177-4206.
>
> [4] Li, Chao, et al. "Dof: A diffusion factorization framework for offline multi-agent reinforcement learning." The Thirteenth International Conference on Learning Representations. 2025.

---

> ### Author Response · Authors · 2025-11-24
>
> **R2. SMAC-v2 (weakness 2, question 2)**
>
> To further validate the robustness of MASTARS in more challenging and realistic scenarios, we have added new experiments on SMAC-v2[5] in the revised version of the paper.SMAC-v2 is substantially more difficult than the original SMAC due to stochastic unit compositions and randomized initial placements for both allied and enemy teams. These characteristics induce highly diverse episode dynamics and make SMAC-v2 a stronger benchmark for assessing multi-agent coordination under distributional variability. Details of the setup and dataset construction are provided in Appendix J (Table 14).
>
> Across both offline RL algorithms considered (CFCQL and OMAR), incorporating trajectories generated by MASTARS consistently yields large performance gains over training on the offline dataset alone. Furthermore, MASTARS demonstrates clear advantages over diffusion- and stitching-based generative baselines (MADiff*, DoF*, MADiTS, and INS), achieving the best or second-best performance on 9/10 tasks under CFCQL and on all 10/10 tasks under OMAR. The complete results are reported in Appendix J (Table 15). The table below provides a compact comparison by the mean performance across all tasks for both CFCQL and OMAR.
>
> | **Method** | &nbsp;&nbsp;Original&nbsp;&nbsp; | &nbsp;&nbsp;MADiFF*&nbsp;&nbsp; | &nbsp;&nbsp;DoF*&nbsp;&nbsp; | &nbsp;&nbsp;MADiTS&nbsp;&nbsp; | &nbsp;&nbsp;INS&nbsp;&nbsp; | &nbsp;&nbsp;Ours&nbsp;&nbsp; |
> |-----------|----------------------------------|---------------------------------|------------------------------|--------------------------------|-----------------------------|------------------------------|
> | **Score** | &nbsp;&nbsp;0.18&nbsp;&nbsp;     | &nbsp;&nbsp;0.21&nbsp;&nbsp;    | &nbsp;&nbsp;0.06&nbsp;&nbsp; | &nbsp;&nbsp;0.02&nbsp;&nbsp;   | &nbsp;&nbsp;0.24&nbsp;&nbsp;| &nbsp;&nbsp;**0.37**&nbsp;&nbsp; |
>
>
> These results confirm that the benefits of MASTARS are not limited to relatively controlled environments such as SMAC and MPE. Even under the highly stochastic and diverse conditions of SMAC-v2, MASTARS remains effective and reliable, indicating strong robustness and scalability to more challenging multi-agent settings.
>
> Regarding the reviewer’s concern about scenarios with a large number of agents, we note that MASTARS was also evaluated on long-horizon (2c\_vs\_64zg) and large-number-of-agent (25m) settings in Appendix G.2 (Scalability Analysis w.r.t.\ Agent Number and Horizon Length). These experiments further show that MASTARS continues to outperform existing generative baselines even as the number of agents and the trajectory horizon grow, demonstrating that the proposed framework scales favorably in both dimensions.
>
> &nbsp;
>
> [5] Ellis, Benjamin, et al. "Smacv2: An improved benchmark for cooperative multi-agent reinforcement learning." Advances in Neural Information Processing Systems 36 (2023): 37567-37593.
>
> &nbsp;
>
> **R3. Ablation of value-ordering (weakness 3, question 3)**
>
> The ablation study for Value-based Agent Ordering is included in our experiments, but it appears in Section 4.6 (Table 5) rather than in Figure 5. Because MASTARS generates agents sequentially by design, it is not meaningful to ablate the ordering component by “removing” it altogether, since removing the ordering would revert the method back to joint generation rather than isolating the effect of ordering itself.
>
> Instead, we ablate the*choice of ordering* by comparing our value-based ordering against two alternatives: (i) random ordering and (ii) reverse ordering (i.e., generating high-value agents last). The results in Table 5 show that random ordering leads to degraded performance, and reverse ordering further reduces coordination quality.
>
> These findings confirm that generating low-value agents last is the most effective strategy, validating the contribution of our value-based ordering mechanism.

---

> > ### Comment · Reviewer_Q8n9 · 2025-11-27
> >
> > Thank you for your response. However, I still have the following questions:
> >
> > 1. If this work is closer to MADiTS and INS, why does Figure 1 compare only with MADiff instead of showing comparisons with both MADiTS and INS?
> >
> > 2. Although MASTARS indeed achieves strong performance compared to the baselines on SMACv2, its variance appears to be relatively large.
> >
> > 3. Could you provide the standard deviation in Table 5?

---

> ### Author Response · Authors · 2025-11-28
>
> We are grateful for the reviewer’s follow-up questions and their recognition of our earlier revisions. We address the remaining concerns as follows.
>
> &nbsp;
>
> **Visual Comparison between prior methods (Question 1)**
>
> The reason we included MADiFF* in the visualization of the main paper is that MADiTS, INS, and MADiFF* all operate as joint generation methods, and for simplicity we chose to illustrate MADiFF* as the representative joint-generation baseline used in our primary experiments.
>
> To directly address the reviewer’s question, we additionally visualize trajectories generated by MADiTS from the same initial state as our method. The results are provided in Appendix L.
>
> As shown in Figure 11 batch 0 and 1, MADiTS frequently produces suboptimal trajectories in which two agents collapse to the same destination, whereas our method consistently generates trajectories in which agents correctly separate toward their respective optimal targets.
>
> Furthermore, MADiTS frequently struggles to generate correct destination points (indicated by 'X' in the figure) and occasionally even fails to produce a feasible trajectory at all. For example, in Batch 2, the blue agent’s trajectory becomes invalid, while our method consistently generates clear and well-separated targets. These issues arise from limitations in its generation procedure. Because MADiTS performs joint conditional diffusion in a single forward pass, it cannot refine or correct individual agent behaviors, which often leads to noisy, inconsistent, or poorly coordinated destination outputs.
>
> In contrast, MASTARS uses a RePaint style inpainting procedure. By iteratively resampling masked portions conditioned on neighboring agents and earlier denoising updates, our method gradually resolves local inconsistencies and forms the trajectory in a coordinated way. This iterative process enables MASTARS to produce reliable, coherent, and well structured trajectories, which MADiTS often fails to achieve.
>
> Separately, a comparison with INS is challenging, because INS generates transitions $(s, a, r, s')$ rather than full trajectories. This is also consistent with the authors’ own clarification in Appendix D of INS[1], where they state that *“MTDiff-S generates data at the trajectory level for unknown tasks, focusing on multi-task scenarios, which is why it is not included as a baseline in our work.”* Accordingly, we restrict our qualitative visualization to MADiTS and our method.
>
> &nbsp;
>
> [1] Fu, Yuqian, et al. "INS: Interaction-aware Synthesis to Enhance Offline Multi-agent Reinforcement Learning." The Thirteenth International Conference on Learning Representations.
>
> &nbsp;
>
> **SMACv2 Variance (Question 2)**
>
> We agree that the variance on SMACv2 is larger than in other settings. However, as noted in the paper, SMACv2 is substantially more stochastic due to varying unit compositions and randomized starting positions. In the offline MARL setting, this inherent stochasticity naturally leads to larger performance variance, and this effect is observed not only in our method but also across other baselines.
>
> It is also important to note that baselines exhibiting smaller variance typically do so because their overall performance is very low (often close to zero), which compresses the range of achievable scores. In contrast, our method shows higher variance precisely because it succeeds on a subset of challenging SMACv2 tasks, resulting in a wider spread of returns. Other baselines that achieve reasonably good performance also display substantial variability, such as INS (0.4$\pm$0.52) under CFCQL on Protoss 3\_vs\_3-Good and MADiFF* (0.48 $\pm$ 0,48) under OMAR on Terran 3\_vs\_3-Medium. Thus, the larger variance reflects the fact that MASTARS is able to solve tasks that other methods fail to address.
>
> &nbsp;
>
> **Table 5 Variance (Question 3)**
>
> Since standard deviations cannot be meaningfully averaged across tasks, we did not report them in tables that present only average performance. For Table 5, we instead provide the per-task standard deviations separately, and these values are included in Appendix K.
>
> &nbsp;
> &nbsp;
>
> We thank the reviewer for their insightful questions, which have helped us improve the clarity and quality of our paper. If there are any further questions or concerns, we would be happy to address them.

---

### Official Review · Reviewer_HnP9 · 2025-10-29

**Soundness:** 3
**Presentation:** 3
**Contribution:** 2
**Rating:** 4
**Confidence:** 4

**Summary:**

The paper proposes MASTARS, a diffusion-based framework designed to augment MARL trajectories.
The method seeks to enhance offline MARL performance by addressing issues related to data quality and coordination among agents.
It achieves this by leveraging a sequential agent-wise generation strategy with return-conditioned subgoals.
The authors demonstrate that their method provides coordinated, diverse, and realistic multi-agent trajectories that improve the performance of offline RL methods when applied to augmented datasets.

**Strengths:**

1. The sequential generation with subgoals provides a new way to generate coordinated behaviors across agents.
2. MASTARS shows consistent improvements over several baseline approaches such as DoF, MADiff, and INS, et al.
3. The paper provides a thorough explanation of the proposed method, making it easy to follow.

**Weaknesses:**

1. The environments (both MPE and SMAC) are relatively simple and may not fully showcase the potential of MASTARS in more complex, real-world, or large-scale settings. Although existing offline MARL methods typically choose these two environments (perhaps for data convenience), this undoubtedly stagnates the community. I believe that tasks in SMAC and MPE do not sufficiently support the claim of "combinatorial complexity in joint modeling." If the authors could provide experiments in more modern or challenging environments, it would significantly enhance the quality of the paper. I am open to revising my rating should such additions be included.
2. The sequential paradigm is a common approach in MARL, yet the related work section lacks coverage of key works in this area, such as MAT[1], PMAT[2], HARL[3], SeqComm[4], DIMA[5], and SeqWM[6].

[1] Multi-Agent Reinforcement Learning is a Sequence Modeling Problem, NeurIPS, 2022.

[2] PMAT: Optimizing Action Generation Order in Multi-Agent Reinforcement Learning, AAMAS, 2025.

[3] Heterogeneous-Agent Reinforcement Learning, JMLR, 2024.

[4] Multi-Agent Coordination via Multi-Level Communication, NeurIPS, 2024.

[5] Revisiting Multi-Agent World Modeling from a Diffusion-Inspired Perspective, arxiv, 2025.

[6] Empowering Multi-Robot Cooperation via Sequential World Models, arxiv, 2025.

**Questions:**

1. In Table 1, the performance of the "medium" setting is sometime worse than the "random" setting. Similarly, Table 2 shows cases where "good" performs worse than "medium." Could the authors clarify the reasons behind these results?
2. Would the concept of subgoals be applicable in environments that are not fully cooperative, or even in competitive settings?

---

> ### Author Response · Authors · 2025-11-24
>
> We thank the reviewer for the insightful comments and hope that our response together with the revised paper satisfies all concerns of the reviewer.  We believe we indeed proposed an efficient way to data augmentation for offline multi-agent RL. The numbers of equations, figures, tables and sections below refer to those in the revised paper available at openreview unless we give special remarks. Our response to each comment are presented below.
>
> &nbsp;
>
> **R1. Modern Challenging environmens (weakness 1)**
>
> To further assess the applicability of MASTARS beyond the controlled scenarios examined in the main experiments, we evaluated our method on SMAC-v2[1], which poses a harder challenge than the original SMAC. In SMAC-v2, both allied and enemy unit compositions, as well as initial agent placements, are sampled from probability distributions rather than fixed configurations. This results in highly stochastic and diverse episodes, making SMAC-v2 a stronger testbed for evaluating generalization in multi-agent coordination. Details of the environment setup and dataset construction are reported in Appendix J (Table 14).
>
> We briefly highlight the key observations as follows. Across both offline RL algorithms considered (CFCQL and OMAR), adding trajectories generated by MASTARS leads to consistent and substantial performance improvements over training with the offline dataset alone. Moreover, MASTARS exhibits clear advantages over diffusion- and stitching-based generative baselines (MADiff*, DoF*, MADiTS, and INS), achieving the best or second-best performance on 9/10 tasks under CFCQL and on all 10/10 tasks under OMAR, while the table below offers a concise comparison by reporting the total average performance across all tasks, averaged over both CFCQL and OMAR.
>
> | **Method** |   &nbsp; Original &nbsp;   |   MADiFF* &nbsp;   |   DoF* &nbsp;  |   MADiTS &nbsp;  | &nbsp;  INS  &nbsp; |  &nbsp; Ours   |
> |-----------|--------------|-------------|----------|------------|---------|----------|
> | **Score** |   &nbsp; 0.18      |   &nbsp;  0.21     |    0.06   | &nbsp;   0.02     |   &nbsp; 0.24 &nbsp; |  &nbsp; &nbsp;**0.37** |
>
> Overall, the SMAC-v2 results confirm that the gains of MASTARS are not restricted to simpler environments such as SMAC and MPE. Even under the stochastic and diverse conditions of SMAC-v2, MASTARS continues to provide reliable performance improvements, demonstrating strong scalability and generalization.
>
> &nbsp;
>
> [1] Ellis, Benjamin, et al. "Smacv2: An improved benchmark for cooperative multi-agent reinforcement learning." Advances in Neural Information Processing Systems 36 (2023): 37567-37593.
>
> &nbsp;
>
> **R2. Adding related works (weakness 2)**
>
> Our original related work section focused on diffusion-based generative models for data augmentation, and thus we did not include methods such as MAT[2], PMAT[3], HARL[4], SeqComm[5], DIMA[6], and SeqWM[7], since they are not generative or diffusion-based approaches.
>
> However, we agree on that these methods are highly relevant in terms of the sequential modeling perspective. Following the reviewer’s suggestion, we have incorporated a discussion of these works into the revised related work section to better position MASTARS within the broader landscape of sequential MARL methods.
>
> &nbsp;
>
> [2] Multi-Agent Reinforcement Learning is a Sequence Modeling Problem, NeurIPS, 2022.
>
> [3] PMAT: Optimizing Action Generation Order in Multi-Agent Reinforcement Learning, AAMAS, 2025.
>
> [4] Heterogeneous-Agent Reinforcement Learning, JMLR, 2024.
>
> [5] Multi-Agent Coordination via Multi-Level Communication, NeurIPS, 2024.
>
> [6] Revisiting Multi-Agent World Modeling from a Diffusion-Inspired Perspective, arxiv, 2025.
>
> [7] Empowering Multi-Robot Cooperation via Sequential World Models, arxiv, 2025.

---

> ### Author Response · Authors · 2025-11-24
>
> **R3. Performance question (question 1)**
>
> The observation that the "medium" setting can perform worse than the "random" setting (and that "good" can underperform "medium") is due to the characteristics of the offline datasets rather than a limitation of our method.
>
> The "medium" and "good" datasets are collected from single policies, resulting in narrow and concentrated state–action distributions. In contrast, the "random" dataset covers a much wider range of the state space, even though its quality is lower. When the trajectory distribution becomes too narrow, offline policies trained on such data may suffer from poor generalization and extrapolation errors, which can lead to lower scores despite coming from "better" demonstrators.
>
> This phenomenon is not specific to MASTARS; the same trend is observed consistently across baselines. For example, in the Simple-Spread task, nearly all algorithms report lower performance under the "medium" setting than under "random," indicating that the issue stems from dataset coverage rather than the augmentation method.
>
> &nbsp;
>
> **R4. Subgoal applicable to competitive environment (question 2)**
>
> The notion of subgoals in MASTARS is not restricted to fully cooperative environments. A subgoal is defined as the state that maximizes an individual agent’s value estimate along its own trajectory, meaning that each agent determines its subgoal solely based on its own value function. Consequently, the mechanism does not rely on full cooperation.
>
> This formulation can naturally extend to partially cooperative or competitive environments. In a competitive setting, an agent would still identify its subgoal by maximizing its own value estimate, which implicitly reflects both its own advantage and the expected behavior of opponents, as captured by the value function. In such cases, the subgoal becomes opponent-aware rather than team-centric.
>
> In the fully cooperative setting, on which we focus in this work, each agent is trained with a value function based on the global reward. Therefore, the subgoal becomes aligned with collective performance, naturally encouraging coordinated observation patterns among agents.

---

> > ### Comment · Reviewer_HnP9 · 2025-11-24
> >
> > Thank you for the authors’ response. The newly added experiments on SMACv2 are informative, and address my earlier concerns. I have therefore decided to raise my score, as I promised in the review. The clarification regarding the performance of “random” and “medium” is also consistent with terminology used in prior published works.
> >
> > Outside the scope of this paper, I would like to share a personal observation: offline MARL remains a highly promising direction, but its progress currently seems to be limited by the lack of convincing datasets.
> > Most existing open-source datasets focus on simplified simulated domains such as SMAC and MPE, where collecting data is relatively easy, which weakens their practical value.
> > If future work could collect and release datasets in real-world multi-robot domains (e.g., outdoor exploration,multi-arm assembly), it would represent a major contribution to the community.

---

> > > ### Author Response · Authors · 2025-11-24
> > >
> > > Dear Reviewer HnP9
> > >
> > > Thank you very much for your positive feedback and raising the score.
> > > We totally agree on that realistic dataset for offline RL is indeed necessary since the goal of offline RL is to avoid costly or risky on-line environment interaction.
> > >
> > > We appreciate your time and efforts
> > >
> > > Authors

---

### Official Review · Reviewer_kcgt · 2025-10-30

**Soundness:** 3
**Presentation:** 2
**Contribution:** 2
**Rating:** 4
**Confidence:** 5

**Summary:**

The authors present MASTARS, a diffusion-driven framework for data augmentation in offline multi-agent reinforcement learning. MASTARS introduces an agent-wise sequential generation mechanism, where each agent’s trajectory is conditioned on those already generated. To further refine the synthetic data, the method incorporates return-conditioned subgoals to selectively reconstruct low-quality segments and applies a value-based ordering scheme that determines which agents are generated first based on their expected returns. By combining these ideas with a diffusion inpainting procedure inspired by image generation, MASTARS achieves better inter-agent coordination without the burden of high-dimensional joint modeling. Empirical results on MPE and SMAC benchmarks indicate that the approach leads to consistent and notable gains over existing offline MARL augmentation baselines.

**Strengths:**

1. The paper is clearly written and logically organized, making it easy to follow the motivation, methodology, and experimental setup. The visualizations effectively support the explanations.

2. The research problem tackled is highly relevant to offline multi-agent reinforcement learning, where coordinating multiple agents from limited or suboptimal data remains an open challenge. The motivation for addressing this issue is well-grounded and convincing.

3. The proposed approach is methodically designed, building upon established principles of diffusion modeling and inpainting while tailoring them to the multi-agent context.

**Weaknesses:**

1. While the paper presents a well-motivated framework, the claimed novelty around the use of diffusion-based inpainting is somewhat limited. Although the authors are the first to explicitly adapt a RePaint-style mechanism to multi-agent data augmentation, similar ideas of trajectory refinement and selective regeneration have already appeared in prior diffusion-based MARL works [1]. As a result, the contribution feels more like a thoughtful integration of existing components, rather than a fundamentally new methodological direction.

2. The paper demonstrates robustness of MASTARS to small random perturbations in value estimates, which is reassuring. That said, offline value estimation is often affected by non-random biases (e.g., limited coverage, distributional shift, or partial observability), not merely random noise. Discussing whether such biases could influence the stability of agent ordering or subgoal selection would strengthen the paper’s practical relevance.

Ref:

[1] Yuan et al. Efficient Multi-agent Offline Coordination via Diffusion-based Trajectory Stitching. ICLR 2025.

**Questions:**

1. Could the authors clarify in what ways MASTARS differs from other offline MARL data augmentation methods, and highlight the relative strengths and weaknesses of these approaches?

2. The paper emphasizes addressing data scarcity in offline MARL via trajectory augmentation. However, all reported experiments combine the augmented trajectories with the original dataset, leaving it unclear what performance can be achieved using the augmented data alone. Could the authors clarify the intended role of the generated trajectories? Are they meant purely as a distributional supplement to the original data, or could they potentially replace sub-optimal original trajectories? Additionally, why is it still necessary to rely on the original sub-optimal dataset if augmentation alone could provide sufficient coverage?

3. In the introduction, the authors note that naively concatenating all agents’ data into a single model for joint generation often leads to sample inefficiency due to high dimensionality. However, in MASTARS, regardless of whether agent-wise or more sophisticated modeling is used, the diffusion model still needs to fit the joint distribution of all agents. Could the authors clarify how their approach actually addresses the high-dimensionality and sample inefficiency issue raised in the introduction?

---

> ### Author Response · Authors · 2025-11-24
>
> We thank the reviewer for the insightful comments and hope that our response together with the revised paper satisfies all concerns of the reviewer. We believe we indeed proposed an efficient way to data augmentation for offline multi-agent RL. The numbers of equations, figures, tables and sections below refer to those in the revised paper available at openreview unless we give special remarks. Our response to each comment are presented below.
>
> &nbsp;
>
> **R1. Comparing with MADiTS and prior works (weakness 1, question 1)**
>
> To the best of our knowledge, among diffusion-based methods applicable to offline MARL data augmentation, only MADiTS[1] and INS[2] explicitly aim to expand logged datasets. MADiff[3] and DoF[4], although diffusion-based, were originally designed to synthesize high-return trajectories and infer actions via inverse dynamics rather than to augment existing data; therefore, we apply only their trajectory generators for a fair comparison. Importantly, all four prior methods (MADiff, DoF, INS, and MADiTS) generate all agents jointly, which requires exploring a trajectory space of size $O(|S|^{N})$, where $S$ is the design space for single agent and $N$ is the number of total agents. This joint sampling becomes difficult to optimize when the offline dataset is limited or suboptimal.
>
> In contrast, MASTARS removes joint sampling entirely. By generating agents sequentially and conditioning each one on previously generated agents, MASTARS reduces the search space to $O(|SN|)$ while still modeling inter-agent dependencies through inpainting mechanism. Although sequential generation increases sampling time, this trade-off is justified by the substantial reduction in the search space.
>
> Regarding MADiTS specifically, the method may initially appear similar because it also resamples particular agents. However, the mechanisms differ fundamentally. MADiTS first samples a joint trajectory, detects underperforming agents, and then performs a single round of conditional resampling for those agent dimensions. Although this selectively updates low-performing agents, the core remains joint diffusion sampling, resulting in the same scalability issues. Furthermore, coordination can be limited since the regenerated agent segments are not progressively adjusted to match the behaviors of the unchanged agents.
>
> MASTARS, on the other hand, constructs trajectories by generating agents one at a time in a sequential manner rather than refining a jointly generated trajectory. The RePaint-style inpainting mechanism repeatedly resamples masked regions conditioned on surrounding agents and past denoising steps, enabling mutual adaptation among agents throughout the generation process rather than only once. Furthermore, subgoal guidance preserves high-quality segments from the dataset while regenerating only necessary parts, and value-based ordering ensures that the agent requiring the greatest flexibility is generated last.
>
> Overall, while MADiTS performs selective refinement of a jointly generated trajectory, MASTARS replaces joint sampling with a scalable agent-wise generation framework that actively promotes inter-agent harmonization and cooperation. This methodology represents a fundamentally different direction rather than an incremental modification of existing approaches.
>
> &nbsp;
>
> [1] Yuan et al. Efficient Multi-agent Offline Coordination via Diffusion-based Trajectory Stitching. ICLR 2025.
>
> [2] Fu, Yuqian, et al. "INS: Interaction-aware Synthesis to Enhance Offline Multi-agent Reinforcement Learning." The Thirteenth International Conference on Learning Representations.
>
> [3] Zhu, Zhengbang, et al. "Madiff: Offline multi-agent learning with diffusion models." Advances in Neural Information Processing Systems 37 (2024): 4177-4206.
>
> [4] Li, Chao, et al. "Dof: A diffusion factorization framework for offline multi-agent reinforcement learning." The Thirteenth International Conference on Learning Representations. 2025.

---

> ### Author Response · Authors · 2025-11-24
>
> **R2. Value function with non-random biases (weakness 2)**
>
> In our setting, the trajectory generation process relies only on states observed in the offline dataset. Since we do not generate trajectories in unseen regions of the state space, biases arising from limited coverage or partial observability do not directly affect the value estimates. Furthermore, as noted in lines 296–300 of the main paper, we train the value function using a SARSA-style update based solely on observed transitions $(s, a, r, s', a')$. This avoids extrapolation and therefore mitigates distributional shift bias.
>
> Even if systematic bias arises in offline value estimation, its impact on MASTARS is limited. Such bias would shift value estimates in an absolute sense, but it would not alter their relative ordering, because all trajectories would be affected similarly. Since both subgoal selection and agent ordering in MASTARS depend only on the relative ranking of values rather than their absolute scale, the mechanism remains stable as long as the ordering between states is preserved. Thus, even under systematic bias, decision consistency is maintained because the method compares value estimates only in a relative manner rather than relying on precise numerical accuracy.
>
> &nbsp;
>
> **R3. Role of generated trajectory (question 2)**
>
> In our framework, the generated trajectories are intended to serve as a distributional supplement rather than a replacement for the original offline dataset. The purpose of augmentation is to enrich the coverage of high-value regions in the state–action space, not to eliminate the original data.
>
> Regarding whether the augmented data alone could be used for training, this is currently not feasible.
> Diffusion generation tends to produce short to medium-length segments reliably, but its expressiveness
> decreases when attempting to generate full trajectories of the same length and diversity as those in the offline dataset. As a result, the augmented dataset alone does not offer the same coverage as the original dataset.
>
> The role of the generated data is therefore complementary: it provides additional high-quality trajectory segments that would be difficult to obtain from the sub-optimal offline data alone, while the original dataset supplies the full-length trajectories and broad state-space coverage needed for stable policy learning. In practice, the combination of both enables the agent to benefit from the improved local optimality of the generated segments without sacrificing the global coverage available in the original dataset.
>
> &nbsp;
>
> **R4. How to address high-dimensionality and sample inefficiency in introduction (question 3)**
>
> MASTARS learns a diffusion model over the joint multi-agent trajectory space during training, since
> coordinated behavior requires modeling inter-agent dependencies. Therefore, the dimensionality of the
> learned distribution is the same as in joint generation.
>
> The key difference arises in the sampling phase rather than in the training phase. Joint sampling requires searching a trajectory design space of size $O(|S|^{N})$ (as mentioned in R1.), which is computationally prohibitive in offline settings with limited suboptimal data. MASTARS avoids this issue by adopting an autoregressive sampling strategy in which agents are generated sequentially. Each agent explores a design space of size $O(|S|)$, yielding an overall sampling complexity of $O(|SN|)$ instead of $O(|S|^{N})$.
>
> In summary, although MASTARS learns the joint distribution to preserve coordination, it does not sample directly from the full high-dimensional joint trajectory space. Coordination instead emerges through sequential inpainting-based generation, which substantially reduces search complexity and mitigates the sample inefficiency issue raised in the introduction.

---

### Official Review · Reviewer_NJRq · 2025-10-31

**Soundness:** 2
**Presentation:** 2
**Contribution:** 2
**Rating:** 4
**Confidence:** 4

**Summary:**

The authors propose a novel diffusion-based framework, MASTARS, for generating coordinated multi-agent trajectories through agent-level sequential generation. MASTARS employs a diffusion inpainting mechanism, where each agent’s trajectory is conditioned on the previously sampled agents’ trajectories. This design enables fine-grained inter-agent coordination while avoiding the complexity of high-dimensional joint modeling.
To further enhance sample quality, MASTARS integrates reward-conditioned subgoals, allowing the model to exploit valuable data that might otherwise be discarded. By combining agent-level generation with goal-conditioned modeling, MASTARS produces realistic and coherent multi-agent rollout trajectories, thereby facilitating more effective offline multi-agent reinforcement learning (MARL) training. Experimental results on benchmark environments demonstrate that MASTARS significantly improves the performance of existing offline MARL algorithms, validating its effectiveness and generality in collaborative multi-agent scenarios.

**Strengths:**

1. This paper introduces a return-conditioned subgoal mechanism that allows the model to selectively regenerate only low-quality segments while preserving high-quality behavior segments from the original dataset. This mechanism improves data quality and generation efficiency, reducing unnecessary or ineffective reconstructions.
2. The method employs a diffusion inpainting mechanism to achieve fine-grained coordination. Experiments (Fig.3 and Fig.6) demonstrate that this strategy effectively generates more coherent and collision-free coordinated behaviors. The experimental validation is comprehensive, showing consistent improvements across algorithms. On the MPE and SMAC multi-agent benchmarks, the approach achieves over 30\% improvement compared to the original datasets.

**Weaknesses:**

1. MADiff is fundamentally an offline multi-agent RL method. Therefore, the characterization on page 2, lines 69–70, describing it as “an existing diffusion model-based data augmentation method” is inaccurate and potentially misleading.
2. The trajectory acceptance criterion relies on a threshold $\epsilon$ (Eq.6); however, the paper does not provide a theoretical justification for the choice of $\epsilon$, nor does it present any sensitivity analysis to assess how varying $\epsilon$ affects the results.
3. Both sub-goal identification and generation ordering rely on the value function. If the value estimates are unstable or corrupted by noise (as shown in Table 4, where performance drops sharply when $\sigma > 0.1$), the quality of the generated trajectories deteriorates significantly.
4. In lines 71–77 on page 2, the author states that “The independent approach frequently …,” and that “the joint method ….” However, in Fig.1, the results of these two generation methods appear to be the opposite.

**Questions:**

1. If trajectory generation is always performed according to descending value estimates, could this potentially over-constrain the behavior of lower-value agents, thereby reducing policy diversity?
2. Is DIRE able to accurately reflect differences in RL trajectory distributions, and is it therefore appropriate to use this metric to assess the novelty of generated samples?
3. MASTARS employs multiple RePaint calls and joint training of multiple models (value, transition, inverse, and reward). Could this approach incur additional inference costs?

---

> ### Author Response · Authors · 2025-11-24
>
> We thank the reviewer for the insightful comments and hope that our response together with the revised paper satisfies all concerns of the reviewer. We believe we indeed proposed an efficient way to data augmentation for offline multi-agent RL.
> The numbers of equations, figures, tables and sections below refer to those in the revised paper available at openreview unless we give special remarks. Our response to each comment are presented below.
>
> &nbsp;
>
> **R1. Misleading in Paper (weakness 1)**
>
> We agree on that our original wording on page 2, lines 69-70 was imprecise. MADiff[1] is indeed proposed as an offline multi-agent RL method rather than a pure "data augmentation method".
>
> In our experiments, however, we  use only the trajectory generation component of MADiff, similar to MADiTS[2], which is also a diffusion-based trajectory generation method. Concretely, MADiff first generates trajectories and then applies its inverse dynamics stage to infer actions from $(s_t, s_{t+1})$ for policy training. In this sense, MADiff effectively functions as a diffusion-based trajectory generator that augments the offline data in our setting.
>
> To avoid confusion, we added a footnote in lines 69-70 clarifying that MADiff is an offline multi-agent RL algorithm and that we use only its trajectory generation module as a baseline, denoted as MADiff$^\ast$.
>
> &nbsp;
>
> [1]Zhu, Zhengbang, et al. "Madiff: Offline multi-agent learning with diffusion models." Advances in Neural Information Processing Systems 37 (2024): 4177-4206.
>
> [2] Yuan et al. Efficient Multi-agent Offline Coordination via Diffusion-based Trajectory Stitching. ICLR 2025.
>
> &nbsp;
>
> **R2. Sensitivity of $\epsilon$ (weakness 2)**
>
> First of all, deriving a theoretical justification for $\epsilon$ is challenging in our setting. To the best of our knowledge, prior methods[2, 4, 5] that adopt conceptually similar transition-based filtering mechanisms likewise do not provide theoretical justification and instead validate the choice of threshold empirically through hyperparameter ablations.
>
> Following this convention, we performed the same type of sensitivity analysis for $\epsilon$, and it was already included in the original submission. Specifically, Appendix G, Figure 8(a) presents the effect of varying $\epsilon$. As shown in the figure, the performance of MASTARS remains stable across a wide range of $\epsilon$ values, indicating that the method is robust to the choice of this hyperparameter.
>
> &nbsp;
>
> [2] Yuan, Lei, et al. "Efficient multi-agent offline coordination via diffusion-based trajectory stitching." The Thirteenth International Conference on Learning Representations. 2025.
>
> [4] Li, Guanghe, et al. "Diffstitch: Boosting offline reinforcement learning with diffusion-based trajectory stitching." arXiv preprint arXiv:2402.02439 (2024).
>
> [5] Lee, Jaewoo, et al. "Gta: Generative trajectory augmentation with guidance for offline reinforcement learning." Advances in Neural Information Processing Systems 37 (2024): 56766-56801.
>
> &nbsp;
>
> **R3. Noise of Value function (weakness 3)**
>
> Table 4 evaluates an intentionally challenging scenario in which we *artificially inject random noise* into the value function itself. This setting does not reflect the behavior of the actual trained value function but is meant to stress-test robustness under extreme perturbations.
>
> In practice, when the value function is trained in our standard SARSA-style manner, relying only on observed transitions $(s, a, r, s', a')$ from the offline dataset. This avoids extrapolation and minimizes estimation error, providing reliable ordering when rewards are noiseless.
>
> As a result, the value estimation error in realistic training conditions is very small (approximately 0.01, as reported in Table 3), and noise levels as large as $\sigma > 0.1$ do not arise in realistic training conditions.

---

> ### Author Response · Authors · 2025-11-24
>
> **R4. Clarification of Figure 1 (weakness 4)**
>
> We apologize for the confusion caused by the caption and thank the reviewer for pointing out this inconsistency. The confusion stems from an earlier figure revision. During manuscript preparation, Figure 1 was updated from a previous version that included a *sequential* generation example at the beginning and did not contain the t-SNE visualization. When replacing the sequential panel with the new t-SNE plot, the caption from the older version was inadvertently retained, resulting in a mismatch between the text description and the final figure.
>
> To clarify:
> - Figure 1(b) corresponds to the **independent** method,
> - Figure 1(c) corresponds to the **joint** method.
>
> This arrangement is consistent with the explanation in the main text. The apparent inconsistency was solely caused by the outdated caption, which has been corrected in the revised manuscript.
>
> &nbsp;
>
> **R5. Sequential sampling vs policy diversity (question 1)**
>
> First, we would like to highlight the advantage of sequential generation compared to joint generation. In principle, generating all $N$ agents jointly would offer maximal flexibility; however, doing so requires exploring a trajectory design space of size $O(|S|^{N})$, where $S$ is the design space for a single agent. This combinatorial explosion makes joint sampling impractical, especially in offline settings with limited sub-optimal data coverage. Our approach adopts a sequential strategy, which reduces the complexity from $O(|S|^{N})$ to $O(|SN|)$ by generating agents one at a time. Although this introduces a sequential structure, it allows the model to capture inter-agent dependencies while remaining computationally tractable.
>
> Moreover, to answer the reviewer's question, the sampling order is not random or fixed, which directly addresses the concern that ordering agents by value might over-constrain those generated later. Instead, agents are generated according to their value estimates, with the lowest-value agent generated last. As described in lines 292–295, low-value agents tend to reach their value peak near the beginning of the segment, meaning that their subgoals are typically reached early and they have greater flexibility in later timesteps. Assigning such agents to the final position in the generation order allows them to explore the remaining design space with minimal influence on previously generated agents, enabling them to adjust and diversify their behavior when necessary. In this way, the sequential sampling procedure maintains policy diversity and coordination flexibility for later agents rather than constraining them.
>
> &nbsp;
>
> **R6. Justification of DIRE (question 2)**
>
> DIRE [1] was originally proposed as a metric for detecting distributional discrepancies between real and generated samples in diffusion-based image generation. Its key idea is to compare the denoising residuals at multiple diffusion timesteps, which serves as a proxy for the distance between underlying data distributions.
>
> Although DIRE was introduced in the context of image generation, the formulation itself is not tied to pixel-space data. The metric can be applied to any data modality as long as a diffusion model is defined for that domain. In our case, trajectories are encoded as vector sequences, and the diffusion model learns their distribution in the same denoising framework used for images.
>
> For this reason, extending DIRE to reinforcement learning is appropriate: it provides a principled way to quantify how different the generated trajectories are from those in the offline dataset. Rather than measuring sample diversity heuristically, DIRE evaluates distributional novelty through the diffusion model’s denoising statistics, which directly reflect the discrepancy between trajectory distributions. Thus, DIRE offers a meaningful and model-consistent measure of novelty in our setting.
>
> [1]Wang, Zhendong, et al. "Dire for diffusion-generated image detection." Proceedings of the IEEE/CVF International Conference on Computer Vision. 2023.
>
> &nbsp;
>
> **R7. Inference cost (question 3)**
>
> First, the additional models used in MASTARS (value, transition, inverse, and reward) are lightweight two-layer MLPs with 256 hidden units (please see Appendix E). Their computational footprint is negligible compared to the diffusion model, so they do not meaningfully increase training or inference time.
>
> Second, RePaint calls introduce some additional inference cost because trajectories are generated sequentially rather than jointly. However, as noted previously in R5, this overhead is the result of a deliberate design choice: sequential sampling avoids the intractable $O(|S|^N)$ complexity of joint trajectory modeling while still capturing inter-agent dependencies, which ultimately leads to better performance.

---

> > ### Comment · Reviewer_NJRq · 2025-11-26
> >
> > Thank you for the authors response. However, the claims are not sufficiently supported by experimental evidence. I will maintain my original score.

---

> > > ### Author Response · Authors · 2025-11-27
> > >
> > > Thank you again for your quick response and for sharing your perspective.
> > > We sincerely appreciate the time and effort you have dedicated to reviewing our work.
> > >
> > > To further improve the quality and clarity of the paper, we would be grateful if you could kindly elaborate on which aspects of the experimental evaluation you find insufficient. Understanding which specific experiments or analyses you believe are missing (or would strengthen the contribution) would help us significantly in preparing a more thorough revision.
> > >
> > > Your guidance would be extremely valuable for us to enhance the paper beyond its current form, and we are fully open to adding additional experiments or analyses that address your concerns.

---

### Official Review · Reviewer_LNsF · 2025-11-01

**Soundness:** 4
**Presentation:** 2
**Contribution:** 2
**Rating:** 4
**Confidence:** 4

**Summary:**

This paper extends data augmentation to multi-agent offline reinforcement learning, identifying and addressing the combinatorial growth of joint state–action spaces and the necessity for coordinated behavior by introducing an inpainting-based diffusion approach. Experimental results on several widely used multi-agent reinforcement learning benchmarks demonstrate that the proposed method achieves a reasonable performance improvement.

**Strengths:**

1. The proposed method is novel. It addresses the combinatorial growth of joint state–action spaces and the necessity for coordinated behavior through the concept of image inpainting, which is a major challenge in data augmentation for multi-agent reinforcement learning.

2. The paper is well-written, logically organized, and methodologically rigorous despite its complexity. It conducts extensive experiments on several popular multi-agent offline benchmarks (such as MPE and SMAC). The results show that MASTARS significantly outperforms existing baselines across various tasks and difficulty levels, demonstrating the effectiveness of the proposed approach.

3. The overall structure of the paper is clear, with well-organized sections introducing each component of the model and the training process.

4. MASTARS effectively tackles the difficulties of data augmentation in multi-agent reinforcement learning caused by joint distribution and cooperation issues. Its  performance on challenging tasks and innovative solutions to complex problems provide valuable insights for the multi-agent reinforcement learning community.

**Weaknesses:**

1. The authors’ explanation of Figure 1 seems problematic. They claim that “The independent approach frequently results in collisions as agents ignore each other’s behavior. In contrast, the joint method improves collision avoidance.” However, based on my observation, in Figure 1(c), the trajectories of the green and red agents also collide—and even intersect more severely than in (b). Doesn’t this contradict the statement that “the joint method improves collision avoidance”?

2. Some symbols are not clearly defined. For example, according to my understanding, in lines 247–248, the statement “The generating reverse sampling inner loop starts with index k = K” refers to K as the number of diffusion steps. However, the “diffusion step K” is only mentioned in the Preliminaries section, and it is not explicitly stated that this notation is consistently used in the Method section. It is recommended to restate the definition of the symbol if it has not been used for a long time.

3. As far as I understand, the value function used to determine the augmentation order in this paper is trained independently for each agent. However, in practice, cooperation among multiple agents is common (e.g., in SMAC). Therefore, using an independently trained value function seems somewhat crude. Since this value function plays a key role in critical stages of the sampling process—such as subgoal selection and generation order decisions. I am concerned that its simplicity may become a bottleneck for the proposed method.

4. During the sampling process, the trajectories of agents are generated sequentially. This implies that the generation of the k(l)-th agent’s trajectory is conditioned on all previously generated trajectories (for k( < l)). I am concerned that as more agents are generated, the decision space for later agents may become increasingly restricted, potentially limiting diversity or coordination flexibility.

5. The early stage of the sampling process depends on subgoals determined by the independently trained value function. As mentioned in Weakness 3, this value function does not consider inter-agent relationships (such as cooperation or competition). This limitation might negatively affect subgoal generation and, consequently, the overall sampling process.

**Questions:**

1. Could the authors provide further clarification regarding the discussion of Figure 1? (Please refer to Weakness 1.)

2. I am concerned that the value function, which does not account for inter-agent relationships (such as cooperation or competition), may affect the decision of the generation order (Section 3.3). (Please refer to Weakness 3, 5.)

3. During the sampling process, each agent’s trajectory is generated sequentially, meaning that the generation of the k(l)-th agent’s trajectory depends on all previously generated trajectories for k < l. Is there a similar mechanism during training to ensure that the diffusion model learns this dependency?

4. I am curious about how this method performs in visual environments.

5. Why does the method generate trajectories sequentially according to the decided order, rather than sampling directly from the joint distribution of all N agents?

---

> ### Author Response · Authors · 2025-11-24
>
> We thank the reviewer for the insightful comments and hope that our response together with the revised paper satisfies all concerns of the reviewer. We believe we indeed proposed an efficient way to data augmentation for offline multi-agent RL. The numbers of equations, figures, tables and sections below refer to those in the revised paper available at openreview unless we give special remarks. Our response to each comment are presented below.
>
> &nbsp;
>
> **R1. Clarification of Figure 1 (weakness 1, question 1)**
>
> We apologize for causing confusion in the description of Figure 1, and and thank you for pointing it out. The issue occurred because, during the paper-writing process, we updated Figure 1 from an earlier version
> (consisting of raw, sequential, independent, and joint without t-SNE) to the current version that includes
> the t-SNE visualization. During this update, a version conflict accidentally overwrote part of the caption,
> resulting in the mismatch you observed.
>
> To clarify:
> - Figure 1(b) corresponds to the **independent** method.
> - Figure 1(c) corresponds to the **joint** method.
>
> This mapping is consistent with the explanation in the main text. The confusion arose solely from the
> caption mismatch, which we have corrected in the revised manuscript.
>
> We appreciate the reviewer’s careful observation. With the corrected caption, the statement in the paper
> that the joint method generally improves collision avoidance compared to the independent method remains
> consistent with the intended visualization and our results.
>
> &nbsp;
>
> **R2. Symbol misunderstanding (weakness 2)**
>
> Thank you for pointing this out. We agree that the symbol $K$, which denotes the number of diffusion
> steps, may be unclear when it reappears in lines 247-248 after not being used for a while. To improve
> clarity, we added a brief reminder in the Method section stating that $K$ is the total number of
> diffusion steps and included a reference to the equation where it is first defined in the
> Preliminaries. This ensures consistent and clear notation throughout the paper.
>
> &nbsp;
>
> **R3. Independent value function (weakness 3, weakness 5, question 2)**
>
> Although we train the value function independently for each agent, the setting we consider is fully cooperative and all agents share the same *global reward*. As a result, each agent's value function is trained to predict the global return and therefore implicitly reflects cooperative structure. Even if the value function is factorized across agents, a high global reward is only achieved when the joint behavior is cooperative, allowing each agent to learn which situations are beneficial for cooperation.
>
> In this sense, the independently trained value function is not purely individualistic; rather, it captures cooperation through the shared reward signal and provides meaningful guidance for both subgoal selection and the generation order in the sampling process.
>
> We agree on that incorporating richer inter agent relationships may further improve performance. Extending
> our method with a value function decomposition approach such as QMIX, or other multi agent value
> factorization techniques, is a promising direction and can be readily integrated into our framework.

---

> ### Author Response · Authors · 2025-11-24
>
> **R4. Sequential Generation vs. Joint Sampling (Weakness 4, question 5)**
>
> Although sequentially generating agents' trajectories may appear to restrict the design space for agents generated later, joint sampling of all $N$ agents is computationally prohibitive. Sampling directly from the joint distribution requires exploring a combinatorial trajectory space of size $O(|S|^{N})$, where $S$ is the design space for a single agent. This exponential growth makes it extremely difficult to identify high quality joint trajectories, especially under limited or sub optimal offline data.
>
> In contrast, our sequentail generation reduces the search complexity from $O(|S|^{N})$ to $O(|SN|)$ by generating one agent at a time. Importantly, coordination among agents is still preserved through our RePaint-style inpainting mechanism, which iteratively resamples masked trajectory segments conditioned on surrounding agents and previously generated samples. This repeated refinement enables agents to update their behaviors in response to one another over multiple denoising cycles, allowing coordinated strategies to emerge despite the sequential generation process.
>
> To further mitigate concerns about reduced flexibility for later agents, our method does not generate agents in a random or fixed order. Instead, we determine the sampling order based on agent values and generate the lowest-value agent last. As noted in lines 292–295, low-value agents tend to exhibit their value peak near the beginning of the segment, meaning that their subgoals are typically resolved early and they require greater flexibility in later timesteps. Placing such an agent at the end of the generation order allows it to utilize the remaining design space more freely rather than being constrained by previously generated agents, enabling it to adjust and diversify its behavior. Consequently, the sequential sampling procedure preserves coordination flexibility and diversity for later agents rather than diminishing their expressiveness.
>
> &nbsp;
>
> **R5. Sequential training (question 3)**
>
> We do not apply the sequential mechanism during training. Similar to the repaint style approaches[1], our method introduces the sequential dependency only at sampling time, not during the diffusion model training. This design allows the diffusion model to remain fully general, without being tied to a particular generation order.
>
> Because the model is trained independently of any ordering, we can flexibly apply different agent orders, subgoals, or value based ordering strategies at sampling time without additional training. This provides a key advantage of our framework, enabling diverse and controllable trajectory generation while keeping the training process simple and broadly compatible with various diffusion model backbones.
>
> [1] Lugmayr, Andreas, et al. "Repaint: Inpainting using denoising diffusion probabilistic models." Proceedings of the IEEE/CVF conference on computer vision and pattern recognition. 2022.
>
> &nbsp;
>
> **R6. Visual environment (question 4)**
>
> Applying our method directly to visual environments is challenging. Although diffusion models are effective for generating single images, our setting requires generating an entire trajectory, which corresponds to a sequence of high dimensional images. For example, while typical image diffusion models generate images of size $256 \times 256$, applying diffusion to a visual trajectory would require generating tensors of size $T \times 256 \times 256$, where $T$ is the trajectory length, while also ensuring that each generated frame is consistent with the transition probabilities of the MDP in a single agent RL setting. This greatly expands the generation space and makes the diffusion process substantially more difficult.
>
> Extending this to the multi agent case is even more challenging because we consider a multi agent system in a partially observable environment (POMDP). This requires generating trajectories for $N$ agents while also ensuring temporal consistency and adherence to the transition probabilities of the MDP in a partially observable multi agent setting, which further increases the complexity.
>
> For these reasons, applying our framework to raw pixel observations is non-trivial. Extending the method to visual domains, potentially through large vision language models or alternative sequential image generators, is an interesting direction for future work. Thank you.

---

> > ### Comment · Reviewer_LNsF · 2025-11-25
> >
> > I greatly appreciate the authors’ detailed response and the effort they have put in. Most of my concerns have been addressed. However, I am still curious about the following questions and concerns :
> >
> > The authors mentioned "low-value agents tend to exhibit their value peak near the beginning of the segment". How do you quantify or identify value peaks?
> >
> > This statement seems to conflict with what is described in the paper. Based on my humble understanding, the authors intend to prioritize agents whose value peaks occur later—that is, the position of the peak should determine the ranking. However, in Equation 7, it can be seen that the paper strictly uses the value of the trajectory’s final observation as the basis for ranking.
> >
> >
> > The heuristic generation method based on value ranking is, in fact, a rather coarse and harsh competition mechanism. Agents with higher values are given higher priority in generating trajectories, while relatively lower-value agents can only generate trajectories in coordination with the preceding agents. As a result, agents ranked later (even if their value is not much lower than that of the highest) value agent—cannot fully expand due to the strict sequential generation order. This can cause sequential generation to perform significantly worse than joint generation in certain cases.
> >
> > The current method is limited to fully cooperative multi-agent reinforcement learning, and its core heuristic (value-based ranking) is designed specifically for full cooperation. Moreover, the RePaint mechanism is also tailored for cooperative settings. I am concerned about the extensibility of this method to other types of agent relations, such as competitive or zero-sum games.
> >
> >
> > Typo in line 248: "equation equation 1".

---

> ### Author Response · Authors · 2025-11-28
>
> Dear Reviewer LNsF
>
> Thanks for further comments and questions.
>
> &nbsp;
>
> **How to identify the peak**
>
> For each agent $i$,  we design a state value function $V_{\phi^i}(s)$ parameterized by $\phi^i$ and learn $V_{\phi^i}(s)$ using a SARSA-style update with the entire offline dataset.  Using the learned state-value function, we determine the peak index for a given trajectory segment. Specifically, for a given trajectory $\tau=[s_1,a_1,s_2,a_2,...,s_H]$, we evaluate $V_{\phi^i}(s)$ for each $t=1,...,H$, and choose the index with the largest value.
>
> &nbsp;
>
> **Reduced flexibility of later-generated agents and how to deal with this**
>
> As noted by the reviewer, our ordering goal is to assign an earlier generation order to agents whose peak occurs later in the trajectory. The reason for this is because we reuse the good portion of each trajectory. For a given agent’s trajectory, when the state value increases over the time steps, we assume the agent is performing a good sequence of actions that lead to progressively better states. Therefore, this value-increasing trajectory segment is considered the “good” portion that we want to preserve. We only regenerate the value-decreasing portion after the peak, because this is where trajectory improvement results from.
>
> As the reviewer pointed out, agents generated later in the sequence must adapt to the previously generated ones. So, it has less room. However, by assigning a later generation order to agents whose value peak occurs earlier, we allow later-generated agent to have a larger design space for generation. This ordering scheme helps mitigate the reduced flexibility or design space for later-generated agents. In fact, mitigating this reduced design space is the main motivation behind our ordering strategy, which is to ensure that later-generated agents are given a sufficiently large design space.

---

> ### Author Response · Authors · 2025-11-28
>
> **Why we think low final values imply early value peaks**
>
> As the reviewer mentioned, this approach is heuristic and based on our intuition. Note that the generation segment cannot be very long, as in many other data augmentation methods. This is because other quantities, such as observations, should also be generated. If the segment is too long, errors may accumulate. Considering the generation segment length is not too long compared with the length of a whole episode, we conjecture that when an agent performs a good sequence of actions throughout the segment, it reaches progressively better states during that segment, the state value increases over time, the peak occurs near the end, and the last value is high. Conversely, if the agent performs a poor sequence of actions, it reaches worse states as the segment progresses, the state value decreases, the peak occurs near the beginning, and the last  value is low. Of course, this reasoning is heuristic.
>
> In the revised paper, we directly adopt the reviewer-suggested ordering scheme based on the value peak index, which eliminates the need for the heuristic reasoning described above (see Section 3.3). This also simplifies the argument and aligns well with our goal of providing more design space to agents generated later. Considering all of the above, we believe that value-based ordering is reasonable, especially when the good portion of the data segment is reused.
>
> We conducted experiments using this new ordering scheme. The average performance across all tasks and algorithms on the MPE and SMAC benchmarks is presented in the table below.
>
> | Task &nbsp; |  &nbsp; Original &nbsp; | &nbsp; MADiFF* &nbsp; | &nbsp; DoF* &nbsp; | &nbsp; MADiTS &nbsp; | &nbsp; INS &nbsp; | &nbsp; Ours-LOV &nbsp; | &nbsp; Ours-SG &nbsp; |
> |:-------:|:-------:|:--------:|:-----:|:-------:|:----:|:---------:|:--------:|
> | **MPE**  | 30.4 | 36.7 | 28.0 | 32.8 | 33.1 | 40.5 | **41.1** |
> | **SMAC** | 0.62 | 0.67 | 0.72 | 0.64 | 0.73 | **0.81** | 0.75 |
>
> As shown in the table, the original ordering scheme, LOV (Last-value-based), and the new ordering scheme, SG (Peak index-based), exhibit similar performance. Compared with other enhancement baselines (MADiFF*, DoF*, MADiTS, and INS), our methods consistently achieve superior results. The complete results are presented in Tables 1 and 2 in the revised paper.
>
> We sincerely appreciate the reviewer’s constructive feedback, which has helped strengthen both the explanation and the evaluation of our generation ordering strategy.
>
> &nbsp;
>
> **Beyond Cooperative Tasks**
>
> We agree with the reviewer that MASTARS is designed for cooperative multi-agent tasks. The value-based ordering and the RePaint mechanism inherently assume shared goals and aligned incentives. Extending our framework to competitive or zero-sum environments would require additional mechanisms that explicitly model opponent interactions. However, since many real-world applications (e.g., autonomous driving, traffic control, smart grids, and multi-robot systems) are inherently cooperative, we believe that focusing on fully cooperative settings still yields methods with broad relevance and impact.
>
> &nbsp;
>
> We hope this satisfies the reviewer's concerns. If there remains further concerns, please let us know.
> We appreciate the reviewer’s insightful comments.

---

### Author Response · Authors · 2025-11-24
**COMMON RESPONSE**

Dear Reviewers

We sincerely thank all reviewers for their valuable comments. The feedback has greatly improved the manuscript and guided important revisions.

We have uploaded a revised version at openreview, and all modified contents are highlighted in blue for ease of reference. Please note that the numbers of equations, figures, tables and sections below refer to those in the revised paper available at openreview now.

---

### Author Response · Authors · 2025-11-30

Dear Area Chair,

Thank you for serving the ML community in this difficult time.
Below we summarize the primary reviewer comments and our response/revision.

MASTARS is a diffusion-based framework that generates coordinated multi-agent trajectories through agent-wise sequential generation with inpainting. By leveraging return-conditioned subgoals, it improves data quality and significantly boosts offline MARL performance.

&nbsp;
&nbsp;

**1. Clarified presentation and figure accuracy**

Reviewers *LNsF* and *NJRq* noted that Figure 1’s caption conflicted with the updated visualization. This was caused by an accidental overwrite during paper revision. We corrected the caption and narrative so the example accurately reflects the benefits of coordinated generation. We also clarified the reuse of the diffusion step notation (K), improving readability in the Method section.

&nbsp;


**2. Reliable value-guided ordering for cooperation**

Reviewers *LNsF*, *kcgt* and *NJRq* questioned whether independent per-agent value functions capture cooperation and remain stable under noise. We clarified that all agents share the global reward, so cooperative behavior is implicitly learned. We also emphasized that the noise test in Table 4 represents an extreme adversarial case; in realistic SARSA-style training, estimation noise remains negligible.

Following Reviewer LNsF’s suggestion, we additionally adopted a peak-index based ordering strategy and gives later-generated agents more flexibility where improvement is needed. The results denoted by “Ours-SG” in Tables 1 and 2 of the main paper show comparable or improved performance, confirming robustness of our value-guided sampling approach.


&nbsp;

**3. Advantage over sequential generation design**

All reviewers raised concerns that sequential generation could restrict the flexibility available to subsequent agents, and they asked for clarification on the advantages of a sequential design. We clarified the central motivation: sampling all agents jointly requires exploring a combinatorial trajectory space of size **$(O(|S|^N))$**, which is computationally intractable in offline MARL. Our method reduces this complexity to **$(O(|S| \cdot N))$** by generating one agent at a time while still maintaining coordination through RePaint-style iterative refinement.

Additionally, *later generation is assigned to low-value / early-peak agents*, ensuring they retain the largest behavioral freedom where improvement is most needed. We also clarified that sequential dependency applies only during sampling, while training remains general and order-agnostic. We explicitly state our focus on  **cooperative tasks**.

&nbsp;

**4. Reinforced empirical validation**

Reviewer *NJRq* requested further justification of ordering choices, trajectory filtering thresholds, and novelty metrics. We already included the sensitivity experiments in Appendix G, showing *stable improvements* across hyperparameter variations, and clarified why **DIRE** remains appropriate for diffusion-based trajectory comparison in vector trajectory spaces.

&nbsp;

**5. Scope of evaluation and extension to more challenging domains (SMACv2)**

While our method already improves offline MARL on widely used cooperative benchmarks (MPE and SMAC), reviewers *HnP9*, and *Q8n9* also asked about broader generalization potential. We now explicitly discuss extension to **more stochastic and harder settings such as SMACv2**, where coordinated multi-agent behavior and partial observability are even more challenging. Moving into SMACv2-style domains is a natural future step and aligns well with our design choice to model inter-agent trajectory dependencies.

&nbsp;
&nbsp;


Overall, the revised paper improves clarity, provides deeper architectural justification, and reinforces empirical robustness. We believe all major concerns have been thoroughly addressed, and the contribution remains strong and impactful for coordinated offline multi-agent data augmentation.

At the time of reversion, we were in productive second-round discussions with Reviewers LNsF and Q8n9, and Reviewer Hnpq agreed to raise their score to 6 based on the revisions.

&nbsp;

Thank you very much.

&nbsp;
&nbsp;

Sincerely,
Authors of 17646

---

### Meta-Review · Area_Chair_onVp · 2026-01-08

**Summary:**

This paper proposes MASTARS, a diffusion-based data augmentation framework for offline multi-agent RL that generates coordinated multi-agent trajectories via agent-wise sequential inpainting with return-conditioned subgoals, achieving more realistic rollouts and improved performance on standard benchmarks. From my reading of this paper and the reviews, this paper is imperfect on clarity and rigor, with inconsistent claims versus figures, unclear notation, and limited justification for key design choices. It is also imperfect on methodological depth and evaluation, as the novelty is incremental, coordination relies on simplistic value functions, and experiments are restricted to relatively simple environments with incomplete ablations. The author-review discussions are somewhat constructive, though some of the reviewers are inactive. Finally I think this paper is not ready for publication in ICLR this year.

**Reviewer Concerns:**

See above text.

**Reviewer Scores:**

Reviewer LNsF may consider improve the score if he saw the rebuttal.

---

### Decision · Program_Chairs · 2026-01-26

Reject